# Composition and light absorption of N-containing aromatic compounds in organic aerosols from laboratory biomass burning

Mingjie Xie[1,2,3,4,*], Xi Chen[4], Michael D. Hays[4], Amara L. Holder[4]

[1]Collaborative Innovation Center of Atmospheric Environment and Equipment Technology, Jiangsu Key Laboratory of Atmospheric Environment Monitoring and Pollution Control, School of Environmental Science and Engineering, Nanjing University of Information Science & Technology, 219 Ningliu Road, Nanjing 210044, China
[2]State Key Laboratory of Pollution Control and Resource Reuse, School of the Environment, Nanjing University, Nanjing, China
[3]Oak Ridge Institute for Science and Education (ORISE), Office of Research and Development, U.S. Environmental Protection Agency, 109 T.W. Alexander Drive, Research Triangle Pak, NC 27711, USA
[4]National Risk Management Research Laboratory, Office of Research and Development, U.S. Environmental Protection Agency, 109 T.W. Alexander Drive, Research Triangle Pak, NC 27711, USA

*Correspondence to: Mingjie Xie

E-mail: mingjie.xie@colorado.edu; mingjie.xie@nuist.edu.cn;

Tel: +86-18851903788;

Fax: +86-25-58731051;

Mailing address: 219 Ningliu Road, Nanjing, Jiangsu, 210044, China

**ABSTRACT**

This study seeks to understand the compositional details of N-containing aromatic compounds (NACs) emitted during biomass burning (BB) and their contribution to light-absorbing organic carbon (OC), also termed brown carbon (BrC). Three laboratory BB experiments were conducted with two U.S. pine forest understory fuels typical of those consumed during prescribed fires. During the experiments, submicron aerosol particles were collected on filter media and subsequently extracted with methanol and examined for their optical and chemical properties. Significant correlations ($p < 0.05$) were observed between BrC absorption and elemental carbon (EC)/OC ratios for individual burns data. However, the pooled experimental data indicated that EC/OC alone cannot explain the BB BrC absorption. Fourteen NAC formulas were identified in the BB samples, most of which were also observed in simulated secondary organic aerosol (SOA) from photo-oxidation of aromatic VOCs with $NO_X$. However, the molecular structures associated with the identical NAC formula from BB and SOA are different. In this work, the identified NACs from BB are featured by methoxy and cyanate groups, and are predominately generated during the flaming phase. The mass concentrations of identified NACs were quantified using authentic and surrogate standards, and their contributions to bulk light absorption of solvent extractable OC were also calculated. The contributions of identified NACs to organic matter (OM) and BrC absorption were significantly higher in flaming-phase samples than those in smoldering-phase samples, and correlated with EC/OC ratio ($p < 0.05$) for both individual burns and pooled experimental data, indicating that the formation of NACs from BB largely depends on burn conditions. The average contributions of identified NACs to overall BrC absorption at 365 nm ranged from $0.087 \pm 0.024$ to $1.22 \pm 0.54\%$, $3 - 10$ times higher than their mass contributions to OM ($0.023 \pm 0.0089$ to $0.18 \pm 0.067\%$), so the

NACs with light absorption identified in this work from BB are likely strong BrC chromophores.
Further studies are warranted to identify more light-absorbing compounds to explain the
unknown fraction (> 98%) of BB BrC absorption.

## 1 Introduction

Biomass burning (BB), including residential burning for cooking, heating, and open
burning, is a major source of atmospheric carbonaceous aerosol, contributing 62% and 93% of
black carbon (BC) and primary organic carbon (OC) particle emissions, respectively (Bond et al.,
2004). BC can absorb sunlight across the entire spectral range with a weak dependence on
wavelength ($\lambda$) (Bond, 2001;Bond et al., 2013;Lack and Langridge, 2013). OC in particulate
matter (PM) is commonly treated as purely light scattering component in global climate models
(Chung et al., 2002;Myhre et al., 2013). Recent field and laboratory studies found that the light
absorption of BB OC increases rapidly from the purple-green region (400–550 nm) to near
ultraviolet (UV) region (300–400 nm) (Kirchstetter et al., 2004;Laskin et al., 2015;Chakrabarty
et al., 2016;Xie et al., 2017b). The light absorption and scattering by BC and OC from BB can
directly affect the Earth's radiative balance (Ramanathan et al., 2001;Anderson et al., 2003;Bond
and Bergstrom, 2006), and BC emission factors and its warming effect have been intensively
investigated (Bond et al., 2004;Bond et al., 2013). However, the optical properties and chemical
composition of light-absorbing OC, also termed brown carbon (BrC) from BB is less well
characterized. The chromophores in BrC are expected to have high degree of unsaturation or
conjugation (Chen and Bond, 2010;Lin et al., 2014;Laskin et al., 2015), but are seldom identified
and used as BrC tracers in the atmosphere (Desyaterik et al., 2013;Zhang et al., 2013;Teich et al.,

75 2016).

Polycylic aromatic hydrocarbons (PAHs) and their derivatives are typical BrC chromophores (Samburova et al., 2016;Huang et al., 2018), of which the light absorption in the UV and visible wavelength range is highly dependent on ring numbers and degree of conjugation (Samburova et al., 2016). However, PAH emissions are not source-specific, but are associated with multiple different combustion processes, including BB (Samburova et al., 2016), coal burning (Chen et al., 2005), motor vehicle emissions (Riddle et al., 2007), etc. Therefore, PAHs are not unique to BB BrC. N-containing aromatic compounds (NACs) are another class of BrC chromophores that have been detected in BB (Lin et al., 2016), cloud water (Desyaterik et al., 2013) and atmospheric particles (Zhang et al., 2013;Teich et al., 2017). In water extracts of atmospheric particles, NACs can contribute greater than 3% of the light absorption at 365–370 nm (Zhang et al., 2013;Teich et al., 2017). These results suggest that NACs are important BrC chromophores, but their composition and structures are less certain for BB aerosols. Nitrophenols, nitrocatechols, and methyl nitrocatechols (including isomers) are commonly observed in BB aerosols (Iinuma at al., 2010; Claeys et al., 2012;Lin et al., 2016;Lin et al., 2017), and are also generated from the photo-oxidation of benzene, toluene, and *m*-creosol in the presence of NO$_X$ (Iinuma et al., 2010;Lin et al., 2015;Xie et al., 2017a). As such, other NAC structures specific to BB are needed to represent BB BrC chromophores. Additionally, very few studies have examined the influence of burn conditions on the formation of NACs in BB emissions, although it is well known that increasing combustion temperature, or flaming dominated combustion, is associated with strong BrC absorption (Chen and Bond, 2010;Saleh et al., 2014).

The present study attempts to characterize the compositional profile of NACs from BB, identify additional NAC structures in laboratory BB samples, and evaluate the contributions of

NACs to bulk absorption of solvent extractable OC from BB. A high-performance liquid
chromatograph interfaced to a diode array detector (HPLC/DAD) and quadrupole (Q)-time-of-
flight mass spectrometer (ToF-MS) was used to examine NACs in $PM_{2.5}$ (particulate matter with
aerodynamic diameter $\leq$ 2.5 μm) from three BB experiments. A thermal-optical instrument
determined bulk OC and elemental carbon (EC) in the PM, and a UV/Vis spectrometer was used
to measure total BrC absorption in methanol extracts of BB $PM_{2.5}$. In this work, a number of
NACs formulas with structures that might be specifically related to BB were identified, and the
contributions of identified NACs to bulk BrC absorption were calculated. These results shed
light on the light-absorbing characteristics of BB OC at bulk chemical and molecular levels,
benefiting the understanding of BrC sources and chromophores.
**2 Methods**
**2.1 Laboratory open BB simulations**
Laboratory simulations of open BB were conducted at the U.S. EPA [Research Triangle
Park (RTP), North Carolina (NC)] Open Burn Test Facility (OBTF), a 70 m$^3$ enclosure, as
detailed in Grandesso et al. (2011). Details of the protocols for biomass fuel collection and burn
simulations were provided elsewhere (Aurell and Gullett, 2013;Aurell et al., 2015;Holder et al.,
2016). Briefly, forest understory fuels were gathered from two different locations in the
southeastern United States — Florida (FL) and NC. The FL forest field (Eglin Air Force Base,
FL) is characteristic of a well-managed long leaf pine (*Pinus palustris*) ecosystem. The NC
forest was located near the EPA campus in RTP, and it contained mainly Loblolly pine (*Pinus
taeda*) with some deciduous hardwood trees leaf litter. Biomass fuel was divided by a quartering
procedure (Aurell and Gullett, 2013) and burned in batches (1 kg) on an aluminum foil-coated
steel pan (1 m × 1 m). Ambient air was pulled into the OBTF through a large inlet at ground

level and the combustion exhaust was drawn through a roof duct near a baghouse using a high-

volume blower. $PM_{2.5}$ was sampled at 10 L min$^{-1}$ on Teflon (47 mm, Pall, Ann Arbor, Michigan,

USA) and pre-heated (550 $^{o}$C, 12 h) quartz filters (QF, diameter 43 mm, Pall) with a $PM_{2.5}$

impactor (SKC, Pittsburgh, Pennsylvania, USA). For the NC forest fire simulation, filter samples

were collected during an initial flaming phase lasting approximately 1–3 minutes. After most of

the flames were extinguished, a second set of filter samples were obtained for the smoldering

emissions. Smoldering samples were collected until there was little or no visible smoke being

emitted from the fuel bed, typically lasting 6–15 minutes. Two separate experiments were done

with the NC forest fuels in spring and summer, respectively, with different ambient temperatures

(Table S1). Sampling of the FL forest fire simulations was done in autumn over the complete

burn, not by combustion phase. Only one experiment was done for the FL forest fuels collected

in fall. Background samples were obtained post-burn inside the OBTF. A summary of the sample

information is provided in Table S1 of the supporting information.

**2.2 Bulk carbon and light absorption measurement**

Details of the bulk OC, EC and light absorption analysis methods are provided in Xie et

al. (2017a,b). Briefly, the bulk OC and EC were measured using an OC-EC analyzer (Sunset

Laboratories, Portland, OR) with a modified NIOSH method 5040 protocol (NIOSH, 1999). For

light absorption measurement, one filter punch (1.5 cm$^{2}$) was extracted in 5 mL methanol (HPLC

grade) ultrasonically for 15 min, and then filtered through a 30 mm diameter

polytetrafluoroethylene (PTFE) filter with a 0.2 μm pore size (National Scientific Company).

The light absorption of methanol extracts was measured with a UV/Vis spectrometer (V660,

Jasco Incorporated, Easton MD) over the wavelength range of 200 to 900 nm. To ensure data

quality, the wavelength accuracy ($\pm$ 0.3 nm) and repeatability ($\pm$ 0.05 nm) were tracked every

month with a NIST traceable Holmium Oxide standard. Solvent background was subtracted with a reference cuvette containing pure methanol. The extracted filter was air dried in a fume hood overnight, and the residual OC was measured with the Sunset thermal-optical analyzer. The extraction efficiency ($\eta$, %) of OC by methanol is calculated by:

$$\eta = \frac{OC_b - OC_r}{OC_b} \times 100\% \qquad (1)$$

where $OC_b$ is the OC content of $PM_{2.5}$ filter before extraction and $OC_r$ is the OC content in the air dried filter after extraction.

The light absorption coefficient of the methanol extracts ($Abs_\lambda$, $Mm^{-1}$) is calculated as:

$$Abs_\lambda = (A_\lambda - A_{700}) \times \frac{V_l}{V_a \times L} \ln(10) \qquad (2)$$

where $A_{700}$ is subtracted from $A_\lambda$ to correct baseline drift, $V_l$ ($m^3$) is the solvent volume (5 mL) used for extraction, $V_a$ ($m^3$) is the air volume of the extracted filter area, $L$ (0.01 m) is the optical path length, and ln (10) converts the absorption coefficient in units of $m^{-1}$ from log base-10 to natural log (Hecobian et al., 2010). The bulk mass absorption coefficient ($MAC_\lambda$, $m^2$ $gC^{-1}$) is calculated by:

$$MAC_\lambda = \frac{Abs_\lambda}{C_{OC}} \qquad (3)$$

where $C_{oc}$ is the mass concentration of extractable OC ($OC_b - OC_r$) for each filter sample ($\mu g\ m^{-3}$). The solution absorption Ångström exponent ($\mathring{A}_{abs}$) is determined from the slope of the linear regression of $\log_{10}(Abs_\lambda)$ vs. $\log_{10}(\lambda)$ over the $\lambda$ range of 300 to 550 nm. In the current work, $Abs_\lambda$ and $MAC_\lambda$ were focused at 365 nm and 550 nm, representing the BrC absorption at near UV and visible regions (Zhang et al., 2013;Saleh et al., 2014), respectively. The EC/OC ratio, methanol extraction efficiency ($\eta$) and light-absorbing properties ($Abs_\lambda$, $MAC_\lambda$ and $\mathring{A}_{abs}$) of each BB sample are listed in Table S1 and summarized in Table 1.

 **2.3 Filter extraction and HPLC/DAD-Q-ToFMS analysis**

168        The $PM_{2.5}$ filter extraction and subsequent instrumental analysis methods used here are

the same as those described in Xie et al. (2017a). Briefly, a 4–6 $cm^2$ piece of each filter was pre-
spiked with 25 μL of 10 ng $μL^{-1}$ nitrophenol-d4 (internal standard, IS), and extracted
ultrasonically in 3–5 mL of methanol twice (15 min each). After filtration and concentration, the
final volume was roughly 500 μL prior to HPLC/DAD-Q-ToFMS analysis. An Agilent 1200
series HPLC equipped with a Zorbax Eclipse Plus C18 column (2.1×100 mm, 1.8 μm particle
size, Agilent Technologies) was used to separate the target NACs with an injection volume of 2
μL. The flow rate of the column was set at 0.2 mL $min^{-1}$, and the gradient separation was
conducted with 0.2% acetic acid (v/v) in water (eluent A) and methanol (eluent B). The
concentration of eluent B was 25% for the first 3 min, increased to 100% from 3 to 10 min, held
at 100% from 10 to 32 min, and then decreased back to 25% from 32 to 37 min. The
identification and quantification of NACs were determined with an Agilent 6520 Q-ToFMS. The
Q-ToFMS was equipped with a multimode ion source operating in electrospray ionization (ESI)
and negative (–) ion modes. All samples were analyzed in full scan mode (40–1000 Da), and an
acceptance criterion of ± 10 ppm mass accuracy was set for compound identification and
quantification. Then selected samples were re-examined using collision-induced dissociation
(CID) technique under identical chromatographic conditions. The MS/MS spectra of target [M–
H]$^-$ ions provided *m/z* data, which was used for identifying NAC structures.

186        The extracted ion chromatograms (EICs) and Q-ToF MS/MS spectra for identified

compounds in selected BB samples are provided in Fig. S1 of the supplementary information and
Fig. 1, respectively. The Q-ToF MS/MS spectra of standard and surrogate compounds used in
this work are obtained from Xie et al. (2017a) and provided in Fig. S2 for comparison. Table 2
provides the formulas, standard/surrogate assignments, and proposed structures of the identified
NACs. Due to the lack of authentic standards, most of the NACs in BB samples were quantified
using surrogates in this work. In general, the surrogate compound with similar molecular weight
(MW) and/or structure was selected for the mass quantification of each identified NAC. Since
the standard compound with hydroxyphenyl cyanate structure is not commercially available,
$C_8H_7NO_4$ and $C_9H_9NO_4$ were quantified as 2-methyl-5-nitrobenzoic acid ($C_8H_7NO_4$) and 2,5-
dimethyl-4-nitrobenzoic acid ($C_9H_9NO_4$), respectively; all the identified NACs with MW > 200
Da were quantified as 2-nitrophloroglucinol ($C_6H_5NO_5$). The mass quantification was conducted
using the internal standard method with 9-point calibration curves (~0.01–2 ng $\mu L^{-1}$). The
compounds corresponding to each NAC formula (including isomers) were quantified
individually and added together for the calculation of mass contribution (%) to organic matter
(OM $\mu g\ m^{-3}$) in each sample. The quality assurance and control (QA/QC) procedures applied for
NACs quantification were provided in Xie et al. (2017a). Field blank and background samples
were free of contamination for NACs. Average recoveries of standard compounds ranged from
75.1 to 116%, and the method detection limit ranged from 0.70 to 17.6 pg (Table S2).
**3 Results and discussion**
**3.1 Light absorption of extractable OC**
The average EC/OC ratio, OC extraction efficiency, $MAC_{365}$, $MAC_{550,}$ and $Å_{abs}$ of all
samples grouped by experiment and fire phase are shown in Table 1. Abbreviations for each
sample group are also listed in the table. The optical properties and bulk composition of the FL
forest samples were reported in Xie et al. (2017b). The average extraction efficiency for all
groups of BB samples is greater than 95% (range 97.0 $\pm$ 1.87 to 99.5 $\pm$ 0.33%), and the light
absorption exhibits strong wavelength dependence, with average $Å_{abs}$ values ranging from 5.68 $\pm$
0.70 to 7.95 $\pm$ 0.22. For each of the two NC forest experiments, the samples collected during the
flaming phase (NF1 and NF2) have significantly higher (student's $t$ test, $p < 0.05$) average
EC/OC ratios, $MAC_{365}$ and $MAC_{550}$, and lower ($p < 0.05$) $\AA_{abs}$ than those collected during the
smoldering phase (NS1 and NS2). When combining the results from the two NC forest
experiments, the average $MAC_{365}$ values for NC forest 2 are significantly ($p < 0.05$) higher than
NC forest 1, despite having a comparable EC/OC ratio (NF1 = 0.042 $\pm$ 0.014 and NF2 = 0.049 $\pm$
0.011, NS1 = 0.0098 $\pm$ 0.0024 and NS2 = 0.0075 $\pm$ 0.0026). Additionally, the average EC/OC
ratio of FF samples is 5–30 times higher than NF and NS samples, while the average $MAC_{365}$
and $MAC_{550}$ values of FF samples (1.13 $\pm$ 0.15 and 0.053 $\pm$ 0.023 $m^2$ $gC^{-1}$) are comparable to
NS1 samples (1.10 $\pm$ 0.11 and 0.054 $\pm$ 0.015 $m^2$ $gC^{-1}$), but lower than other NC forest samples.

223        High temperature pyrolysis or intense flaming conditions are known to increase the

fraction of EC in the total carbonaceous aerosol emissions of BB (Hosseini et al., 2013;Eriksson
et al., 2014;Martinsson et al., 2015;Nielsen et al., 2017). Several studies found that the light-
absorbing properties of BB OC could be parameterized as a function of the EC/OC or
BC/organic aerosol (OA) ratio, a measurement proxy for burn conditions (McMeeking et al.,
2014;Saleh et al., 2014;Lu et al., 2015;Pokhrel et al., 2016), and inferred that the absorptivity of
BB OC depended strongly on burn conditions, not fuel type. In Xie et al. (2017b), significant
correlations ($p < 0.05$) between $MAC_{365}$ of methanol extractable OC from BB and EC/OC ratios
were observed only for samples with identical fuel type, but not for pooled samples with
different fuel types, indicating that both burn conditions and fuel types can impact the light
absorption of BB OC. The contradiction is possibly ascribed to different approaches used in
characterizing the light absorption of BB OC and different test fuel types (Xie et al., 2017b). In
the current work, we combined the sample measurements from all three BB experiments and
analyzed the correlations of bulk $MAC_{365}$ vs. EC/OC. For the analysis, we removed one FL
forest experiment sample due to the extremely high EC/OC ratio of 0.58 (burn 3, Table S1).
Generally, EC/OC ratios are $< 0.4$ for laboratory BB (Akagi et al., 2011;Pokhrel et al., 2016;Xie
et al., 2017b), and $\leq 0.1$ for field BB (Aurell et al., 2015;Xie et al., 2017b;Zhou et al., 2017).
Thus, the burn condition of the FL forest burn 3 (Table S1) is unrepresentative of laboratory BB
simulations or field BB. In Fig. 2a, the bulk $MAC_{365}$ of methanol-extracted OC correlated
significantly ($p < 0.05$) with EC/OC for each BB experiment. However, grouping these sample
measurements resulted in no correlation between $MAC_{365}$ and EC/OC ratio (Fig. 2b). Similar
results were also observed for $MAC_{550}$ vs. EC/OC and $\mathring{A}_{abs}$ vs. EC/OC correlations (Fig. S3a–d).
These results supported that BB BrC absorption depended on more than fire conditions, and
light-absorbing components can be formed at relatively low EC/OC (e.g., tar balls) from
smoldering biomass combustion (Chakrabarty et al., 2010).
In this work, both the comparison of the flaming versus smoldering samples for each
NC experiment (Table 1) and the regressions of bulk $MAC_{365}$ versus EC/OC for individual burns
(Fig. 2a) suggest that the light absorption of OC from BB is strongly dependent on burn
conditions when the fuel type and ambient conditions are similar. The comparison of the FL
versus NC forest experiments (Table 1) and the relationship between bulk $MAC_{365}$ and EC/OC
for grouped measurements (Fig. 2b) indicate that the burn conditions are not the only factor
impacting BB OC absorption. The two NC forest experiments were conducted in spring and
summer, respectively, with distinct ambient conditions (Table S1), and their average $MAC_{365}$
values were significantly ($p < 0.05$) different. This could be partly ascribed to the fact that more
semi-volatile OC (SVOC) will partition into gas phase in summer with higher ambient
temperatures, and the SVOC is less light-absorbing than OC with low volatility (Chen et al.,
2010;Saleh et al., 2014). However, if the relative abundance of EC and OC from BB emissions is
similar between the two NC forest experiments, the evaporation of SVOCs in summer will lead
to higher EC/OC ratios, which is not observed in Table 1. No previous study investigated the
seasonal variation in BrC absorption from BB with similar fuel type. Chen et al. (2001) found
that the ambient temperature might play a role in EC production from traffic by changing the air
density. We suspected that the BB samples from NC forest 2 combustion in summer contained
much stronger light-absorbing components than NC forest 1 combustion in spring, although the
formation mechanism of these strong BrC components is uncertain and merits further study.
Therefore, the light absorption of BB OC is influenced by factors other than burn conditions, and
EC/OC ratios alone may not predict BB OC light absorption from burns with varying fuel types
and ambient conditions.
**3.2 Identification and quantification of NACs**
In the current work, fourteen NAC chemical formulas in BB samples were identified
(Table 2) using the HPLC/DAD-Q-ToFMS analysis, covering all the NACs with high abundance
and strong absorption in ambient and BB particles reported in previous work (Claeys et al.,
2012;Mohr et al., 2013;Zhang et al., 2013;Chow et al., 2016;Lin et al., 2016;Lin et al., 2017).
Their EICs are provided in Fig. S1. The NACs structures corresponding to each chemical
formula were examined using MS/MS data in Fig. 1. In Table S3, the averages and ranges of
relative mass contributions of identified NACs to OM are provided by BB experiment and burn
condition. Here the OM mass was calculated as $1.7 \times$ OC mass (Reff et al., 2009). In addition,
the average relative mass contributions of each NAC in BB samples are shown in Fig. 3.
The three BB experiments have consistent mass contribution profiles (Fig. 3), although
they used different fuel types and were conducted in different seasons. In Table S3, the BB
samples collected during flaming periods (NF1 and NF2) contain significantly higher ($p < 0.05$)
average relative mass contributions from total NACs to OM (tNAC$_{OM}$%: NF1 0.18 $\pm$ 0.067%,
NF2 0.16 $\pm$ 0.045%) than those collected during smoldering periods (NS1 0.055 $\pm$ 0.026%, NS2
0.023 $\pm$ 0.0089%). During the FL forest burn experiment, flaming and smoldering phases were
not separated for sampling, and the average tNAC$_{OM}$% is 0.13 $\pm$ 0.059%, which is between the
tNAC$_{OM}$% of the flaming and smoldering samples of the NC forest experiments. If we
recalculate the average tNAC$_{OM}$% for the NC forest experiments by combining the flaming and
smoldering sample data in each burn, the three BB experiments (FL forest, NC forest 1 and 2)
show similar average tNAC$_{OM}$% (0.11 $\pm$ 0.017–0.13 $\pm$ 0.059%), and the average tNAC$_{OM}$%
across all samples in this work is 0.12 $\pm$ 0.051% (range 0.037 to 0.21%). This value is
comparable to that observed at Detling ($\sim$ 0.5%), United Kingdom during winter, when domestic
wood burning is prevalent (Mohr et al., 2013). In the current work, most of the NACs were
quantified using surrogates, and their contributions to OM from BB may change if authentic
standards or different surrogates are used for quantification. However, the three experiments
might still have consistent relative mass contribution profiles of NACs and similar average
tNAC$_{OM}$%, assuming burn conditions and fuel types have minor impact on the OM/OC ratio. As
shown in Fig. S3e and Fig. 2c, tNAC$_{OM}$% correlated ($p < 0.05$) with EC/OC for both individual
burns and pooled experimental data. Therefore, unlike the light absorption of methanol
extractable OC, the formation of NACs in BB seems to depend largely on burn conditions, rather
than fuel types and ambient conditions.
Among the fourteen identified NAC formulas, $C_6H_5NO_4$ and $C_9H_9NO_4$ have the highest
concentrations (Fig. 3) in FL forest and NC forest flaming-phase samples, accounting for 0.029 $\pm$
0.011 to 0.037 $\pm$ 0.011% and 0.023 $\pm$ 0.012 to 0.049 $\pm$ 0.016% of the OM, respectively (Table
S3). In NC forest smoldering-phase samples, $C_6H_5NO_4$ has the highest mass contribution (NS1
0.024 ±0.0098%, NS2 0.010 ±0.0027%), followed by $C_7H_7NO_4$ (NS1 0.0087 ±0.0030%, NS2
0.0043 ±0.0010%) and $C_9H_9NO_4$ (NS1 0.0052 ±0.0033%, NS2 0.0047 ±0.0013%) (Table S3).
The $C_6H_5NO_4$ was identified as 4-nitrocatechol by comparing its MS/MS spectrum (Fig. 1b) with
that of an authentic standard (Fig. S2b) in Xie et al. (2017a). The EIC of $C_9H_9NO_4$ exhibited 3–4
isomers (Fig. S1i), while only two MS/MS spectra (Fig. 1l,m) were obtained due to the weak
EIC intensity for compounds eluting at times ≥ 10 min. The fragmentation patterns of $C_9H_9NO_4$
compounds (Fig. 1l,m) are different from that of 2,5-dimethyl-4-nitrobenzoic acid (reference
standards with the same formula, Fig. S2g) without the loss of $CO_2$, suggesting that the $C_9H_9NO_4$
compounds identified in this work lack a carboxylic acid group. Both MS/MS spectra of the two
$C_9H_9NO_4$ isomers reflect the loss of OCN (Fig. 1l,m), suggesting a skeleton of benzoxazole/
benzisoxazole or the existence of a cyanate (–O–C≡N) or isocyanate (–N=C=O) group. Volatile
organo-isocyanate structures (e.g., $CH_3NCO$) were identified from anthropogenic biomass
burning (Priestley et al., 2018), and benzoxazole structures have been observed in pyrolyzed
charcoal smoke (Kaal et al., 2008). Giorgi et al. (2004) investigated the fragmentation of 3-
methyl-1,2-benzisoxazole and 2-methyl-1,3-benzoxazole using a CID technique under different
energy frames, and found a loss of CO but not OCN for both of them. In this work, four standard
compounds, including phenyl cyanate ($C_6H_5OCN$), benzoxazole ($C_7H_5NO$), 4-methoxyphenyl
isocyanate ($CH_3OC_6H_4NCO$), and 2,4-dimethoxyphenyl isocyanate  [$(CH_3O)_2C_6H_3NCO$] were
analyzed using a gas chromatography (Agilent 6890) coupled to a mass spectrometer (Agilent
5975B) under electron ionization (EI, 70 ev) mode. These compounds do not have a phenol
structure and cannot be detected using ESI under negative ion mode. The MS/MS spectra of 4-
methoxyphenyl isocyanate and 2,4-dimethoxyphenyl isocyanate were obtained by using a
modified method (ESI at positive ion mode) for NACs analysis in this work. As shown in Fig.
S4a and b, the loss of OCN is observed for phenyl cyanate, but not benzoxazole. In Fig. S4c and
d, the ions at $m/z$ 106 and 136 can be produced from the species at $m/z$ 149 and 179 through the
loss of $CH_3 + CO$ or $H + NCO$ (43 Da). The MS/MS spectra of 4-methoxyphenyl isocyanate and
2,4-dimethoxyphenyl isocyanate (Fig. S4e,f) confirmed the loss of $CH_3 + CO$, and the loss of
$CH_3$ reflected the presence of methoxy group. As such, the $C_9H_9NO_4$ compounds identified in
this work is expected to contain a phenyl cyanate structure.
$C_6H_5NO_3$ (Fig. 1a) is identified as 4-nitrophenol using an authentic standard (Fig. S2a).
$C_7H_7NO_4$ has at least two isomers as shown in Fig. S1c that are identified as 4-methyl-5-
nitrocatechol and 3-methyl-6-nitrocatechol according to Iinuma et al. (2010) and Xie et al.
(2017a). Referring to the MS/MS spectrum of 4-nitrocatechol (Fig. S2b), the $C_6H_5NO_5$
compound should have a nitrocatechol skeleton with an extra hydroxyl group on the benzene
ring. Like $C_9H_9NO_4$ (Fig. 1l,m), the loss of OCN was observed for the fragmentation of
$C_8H_7NO_4$ in the MS/MS spectra (Fig. 1f,g), and a phenyl cyanate structure was proposed (Table
2). However, the fragmentation mechanism associated with the loss of single nitrogen is
unknown and warrants further study. The $C_8H_9NO_4$ identified in this work should have several
isomers (Fig. S1f), and two representative MS/MS spectra are provided in Fig. 1h and i. The first
isomer of $C_8H_9NO_4$ has a dominant ion of $m/z$ 137, reflecting the loss of NO and $CH_3$.
Comparing to the MS/MS spectrum of 4-nitrophenol (Fig. S2a), the first $C_8H_9NO_4$ isomer might
contain a methyl nitrophenol skeleton with a methoxy group. The fragmentation pattern of the
second isomer of $C_8H_9NO_4$ is similar as $C_7H_7NO_4$, and the molecule is postulated as ethyl
nitrocatechol. $C_7H_7NO_5$ has a similar fragmentation pattern as $C_6H_5NO_4$ and $C_7H_7NO_4$, and is
identified as methoxy nitrocatechol. For NC forest burns, $C_{10}H_7NO_3$ was only detected in
flaming-phase samples (Fig. 3). The MS/MS spectrum of $C_{10}H_7NO_3$ was subject to considerable
noise, although the loss of $NO_2$ could be identified (Fig. 1k). In Fig. 1n, the ion at $m/z$ 167 is
attributed to the loss of two $CH_3$ from the [M-H]$^-$ ion of $C_8H_9NO_5$, and the loss of H + CO + NO
is a common feature shared by several nitrophenol-like compounds (Fig. 1b,c,e,i), so the
$C_8H_9NO_5$ compound was identified as dimethoxynitrophenol. The MS/MS spectra of $C_{10}H_{11}NO_4$,
$C_{10}H_{11}NO_5$, $C_{11}H_{13}NO_5$, and $C_{11}H_{13}NO_6$ were characterized by the loss of $CH_3$ and/or OCN (Fig.
1o–t), indicting the existence of methoxy and/or cyanate groups (Fig. S4). Although the exact
structure of these NACs cannot be determined, their functional groups on the benzene ring were
proposed in Table 2 from their fragmentation patterns.
In this work, three of the identified NACs, 4-nitrophenol, 4-nitrocatechol, and methyl
nitrocatechols, were commonly observed in BB emissions or BB impacted atmospheres (Claeys
et al., 2014;Mohr et al., 2013;Budisulistiorini et al., 2017). These compounds can also be
generated from the photo-oxidation of aromatic VOCs in the presence of $NO_X$ (Iinuma et al.,
2010;Lin et al., 2015;Xie et al., 2017a). Both BB and fossil fuel combustion can emit a mixture
of aromatic precursors (e.g., benzene, toluene) for secondary NACs formation (Martins et al.,
2006;Lewis et al., 2013; George et al., 2014;Gilman et al., 2015;Hatch et al., 2015;George et al.,
2015). Therefore, the NACs uniquely related to BB are needed to represent BB emissions. In this
work, the NACs formula with molecular weight (MW) < 200 Da (from $C_6H_5NO_3$, 138 Da to
$C_8H_9NO_5$, 198 Da) were all identified in secondary organic aerosol (SOA) generated from
chamber reactions with $NO_X$ (Xie et al., 2017a). However, the NACs from BB emissions and
SOA formations with identical formulas might have different structures. For example, the
MS/MS spectra of $C_7H_7NO_5$ and $C_8H_9NO_5$ from BB in this work and aromatic VOCs/$NO_X$
reactions in Xie et al. (2017a) had distinct fragmentation patterns (Fig. S5). In Xie et al. (2017a),
the $C_8H_7NO_4$ and $C_9H_9NO_4$ generated from ethylbenzene/$NO_X$ reactions might have fragile
structures and their MS/MS spectra were not available. In this work, $C_8H_7NO_4$ and $C_9H_9NO_4$
from BB emissions are more stable and are supposed to have a phenyl cyanate structure. Among
the four NAC formulas with MW > 200 Da identified in this work (Table 2), $C_{10}H_{11}NO_4$ was
also observed as 5-methoxy-4-nitro-2-(prop-2-en-1-yl)phenol in SOA from reactions of methyl
chavicol and $NO_X$ (Pereira et al. (2015), which cannot be assigned to the $C_{10}H_{11}NO_4$ from BB
emissions in this work. Compared to the NACs in aromatic VOCs/$NO_X$ SOA (Iinuma et al., 2010;
Lin et al., 2015;Xie et al., 2017a; Pereira et al., 2015), the structures of NACs from BB in this
work were characterized by methoxy and cyanate groups. The methoxyphenol structure is a
feature in polar organic compounds from BB (Schauer et al., 2001;Simpson et al.,
2005;Mazzoleni et al., 2007). The cyanate group was rarely reported in gas- or particle-phase
pollutants from BB, which might be a missed feature of BB NACs. Vähä-Savo et al. (2015)
found that cyanate could be formed during the thermal conversion (e.g., pyrolysis, gasification)
of black liquor, which is the waste product from the kraft process when digesting pulpwood into
paper pulp and composed by an aqueous solution of mixed biomass residues. According to Table
2 and Fig. 3, the NACs containing methoxy and/or cyanate groups are predominately generated
during the flaming phase in the two NC forest experiments. Before using these compounds as
source markers for BB NACs, additional work is warranted to understand their exact structures
and lifetimes in the atmosphere. The quantification of these compounds might also be subject to
high variability due to the usage of surrogates.
**3.3 Contribution of NACs to Abs$_{365}$.**

395        For each sample extract, individual NACs contributions to Abs$_{365}$ (Abs$_{365,NAC}$%) were

calculated using their mass concentrations (ng m$^{-3}$) and the MAC$_{365}$ values of individual
compound standards ($MAC_{365,NAC}$), as applied in Zhang et al. (2013) and Xie et al. (2017a). Here,
the $MAC_{365,NAC}$ value is OM based with a unit of $m^2\ g^{-1}$. Each NAC formula was assigned to an
authentic or surrogate standard compound to estimate the contribution to $Abs_{365}$ of extracted OM
(Table 2). Except the NACs with a phenyl cyanate structure, the standard compounds used for
the NACs absorption calculation and mass quantification were the same (Table 2), and their UV-
Vis spectra were obtained from Xie et al. (2017a) and shown in Fig. S6a. The UV-Vis spectra of
three standard compounds with cyanate or isocyate groups are given in Fig. S6b, and none of
them has absorption in the range from 350 to 550 nm. As such, the NACs with cyanate groups
identified in this work were supposed to have no contribution to bulk $Abs_{365}$. Details of the
method for $Abs_{365,NAC}$% calculation are provided in Xie et al. (2017a) and the $MAC_{365,NAC}$ values
for identified NACs formulas in this work are listed in Table S4. Since the standard compounds
used in this work have no absorption at 550 nm, the identified NACs contributions to $Abs_{550}$
were expected to be 0. The average and ranges of $Abs_{365,NAC}$% in BB samples are listed in Table
S5. For simplicity, the average $Abs_{365,NAC}$% in the five groups of BB samples (FF, NF1 and 2,
NS1 and 2) are stacked in Fig. 4.

412       In general, the average contributions of total NACs to $Abs_{365}$ ($Abs_{365,tNAC}$% $0.087\ \pm 0.024$

to $1.22\ \pm 0.54$%) were 3–10 times higher than their average $tNAC_{OM}$% ($0.023\ \pm 0.0089$ to $0.18\ \pm$
$0.067$%) in BB samples (Tables S5 and S3), indicating that the identified NACs with
contributions to $Abs_{365}$ (not including those with cyanate groups) are strong BrC chromophores.
Similar to the NACs mass contributions and compositions, the samples collected during flaming
periods (NF1 and NF2) had significantly higher ($p < 0.05$) average $Abs_{365,tNAC}$% (NF1 $1.21\ \pm$
$0.38$%, NF2 $0.42\ \pm 0.15$%) than those collected during smoldering periods (NS1 $0.72\ \pm 0.27$%,
NS2 $0.087\ \pm 0.024$%); $Abs_{365,tNAC}$% correlated ($p < 0.05$) with EC/OC for both individual burns
(Fig. S3f) and pooled experimental data (Fig. 2d). $C_6H_5NO_4$ (0.037 $\pm$ 0.0080 to 0.31 $\pm$ 0.11%)
and $C_7H_7NO_4$ (0.029 $\pm$ 0.0051 to 0.27 $\pm$ 0.12%) have the highest $Abs_{365,NAC}$% among the
identified NACs across all the three BB experiments (Table S5). The average $Abs_{365,tNAC}$%
values here are comparable to those obtained for atmospheric particles in Germany (0.10 $\pm$ 0.06
to 1.13 $\pm$ 1.03%) (Teich et al., 2017) and Detling, United Kingdom (4 $\pm$ 2%) (Mohr et al., 2013),
but more than 10 times lower than those from chamber reactions of benzene (28.0 $\pm$ 8.86%),
naphthalene (20.3 $\pm$ 8.01%) and *m*-cresol (50.5 $\pm$ 15.8%) with $NO_X$ (Xie et al., 2017a). Lin et al.
(2016, 2017) calculated the absorbance fraction contributed by NACs in BB OC based on signal
peaks at particular retention times in HPLC/photodiode array (PDA) spectrophotometry
chromatograms, and attributed a large portion (up to or greater than 50%) of the solvent extracts
absorption to a limited number of NACs with MW mostly lower than 500 Da. However, the
absorbance signals in HPLC/PDA chromatograms are composed by a mixture of light-absorbing
compounds due to coelution, and some of them are not NACs or even cannot be ionized with ESI.
In this study, standards or surrogates were used to calculate absorption for individual NACs
molecules. These different approaches gave different results. Di Lorenzo et al. (2017) studied the
absorbance as a function of molecular size of organic aerosols from BB, and concluded that the
majority of aqueous extracts absorption ($\lambda$ = 300 nm) was due to compounds with MW greater
than 500 Da and carbon number greater than 20. In this work, less than 2% of the BrC absorption
in BB aerosols at $\lambda$ = 365 was ascribed to the identified NACs with a MW range of 138 to 254
Da, of which the contribution at longer wavelength ($\lambda$ = 550 nm) was expected to be 0. Future
work is needed to identify high MW light-absorbing compounds in BB aerosols to apportion a
greater fraction of BrC absorption in BB aerosols.
**4 Conclusions**
The comparisons of light-absorbing properties ($MAC_{365}$, $MAC_{550}$, and $Å_{abs}$) of BB OC
with EC/OC in this study support that burn conditions are not the only factor impacting BrC
absorption. Other factors like fuel type or ambient conditions may also play important roles in
determining BrC absorption from BB. It may be impractical to predict BrC absorption solely
based on EC/OC ratios in BB emissions from different fuels or over different seasons. The
present study identified fourteen NAC chemical formulas in BB aerosols. The average $tNAC_{OM}$%
of the FL forest, NC forest 1 and 2 (flaming and smoldering samples were combined)
experiments were $0.13 \pm 0.059$%, $0.13 \pm 0.067$%, and $0.11 \pm 0.017$% by weight, respectively,
and the NAC composition was also similar across the three BB experiments. Most of the NACs
formulas identified in this work were also observed in simulated SOA generated from chamber
reactions of aromatic VOCs with $NO_X$, but the same NAC formula from BB and SOA could not
be assigned to the identical compound. In this work, the structures of NACs from BB were
characterized by methoxy and cyanate groups, which were predominately generated during the
flaming phase and might be an important feature for BB NACs. More work is warranted to
understand their exact structures and lifetimes. The average $tNAC_{OM}$% and $Abs_{365,tNAC}$% of the
flaming-phase samples were significantly higher ($p < 0.05$) than those of smoldering-phase
samples in the two NC forest BB experiments. Unlike the bulk $MAC_{365}$ and $MAC_{550}$, $tNAC_{OM}$%
and $Abs_{365,tNAC}$% correlated ($p < 0.05$) with EC/OC for both individual burns and pooled
experimental data, suggesting that burn conditions are an important factor in determining NACs
formation in BB. Except the compounds with cyanate groups, the NACs identified in this work
are likely strong BrC chromophores, as the average contributions of total NACs to bulk $Abs_{365}$
($0.0.087 \pm 0.024$ to $1.22 \pm 0.54$%) are 3−10 times higher than their average mass contributions to
OM ($0.023 \pm 0.0089$ to $0.18 \pm 0.067$%). However, more light-absorbing compounds from BB
with high MW need to be identified to apportion the unknown fraction (> 98%) of BrC
absorption.


**Competing interests**
The authors declare that they have no conflict of interest.
**Disclaimer**
The views expressed in this article are those of the authors and do not necessarily represent the
views or policies of the U.S. Environmental Protection Agency.
**Author contribution**
MX and AH designed the research. MX and XC performed the experiments. AH and MH
managed sample collection. MX analyzed the data and wrote the paper with significant
contributions from all co-authors.
**Acknowledgements**
This research was supported by the National Natural Science Foundation of China (NSFC,
41701551), the State Key Laboratory of Pollution Control and Resource Reuse Foundation (No.
PCRRF17040), and the Startup Foundation for Introducing Talent of NUIST (No.
2243141801001). We would like to acknowledge Brian Gullett, Johanna Aurell, and Brannon
Seay for assistance with laboratory biomass burning sampling. This work was funded by the U.S.
Environmental Protection Agency. Data used in the writing of this manuscript is available at the
U.S. Environmental Protection Agency's Environmental Dataset Gateway (https://edg.epa.gov).

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

**Table 1.** EC/OC ratio, OC extraction efficiency and light-absorbing properties of organic aerosols in PM$_{2.5}$ from laboratory biomass burning.

| Experiment | Phase | Abbr. | Fuels | EC/OC | Extraction efficiency (%) | MAC$_{365}$ (m$^2$ gC$^{-1}$) | MAC$_{550}$ (m$^2$ gC$^{-1}$) | Åabs |
|---|---|---|---|---|---|---|---|---|
| FL forest[a] | No separation | FF | long leaf pine (N=9) | 0.21 ± 0.16 | 97.0 ± 1.87 | 1.13 ± 0.15 | 0.053 ± 0.023 | 7.36 ± 0.59 |
| NC forest 1 | Flaming | NF1 | hardwood/loblolly pine (N=3) | 0.042 ± 0.014 | 97.7 ± 0.41 | 1.47 ± 0.25 | 0.15 ± 0.065 | 5.68 ± 0.70 |
| | Smoldering | NS1 | hardwood/loblolly pine (N=3) | 0.0098 ± 0.0024 | 97.9 ± 0.22 | 1.00 ± 0.11 | 0.054 ± 0.015 | 6.83 ± 0.52 |
| NC forest 2 | Flaming | NF2 | hardwood/loblolly pine (4) | 0.049 ± 0.011 | 99.5 ± 0.33 | 4.07 ± 0.15 | 0.17 ± 0.0051 | 7.38 ± 0.069 |
| | Smoldering | NS2 | hardwood/loblolly pine (4) | 0.0075 ± 0.0026 | 99.2 ± 0.10 | 3.25 ± 0.35 | 0.12 ± 0.033 | 7.95 ± 0.22 |

[a] Data were obtained from Xie et al. (2017b).

**Table 2. Identified N-containing aromatic compounds by HPLC/ESI-Q-ToFMS from laboratory biomass burning in this study.**

| Suggested Formula | Theoretical m/z [M-H]⁻ | Measured m/z [M-H]⁻ | Proposed structure | Quantified as[b] | Absorbing as[c] |
|---|---|---|---|---|---|
| $C_6H_5NO_3$ | 138.0196 | 138.0198 | | 4-Nitrophenol ($C_6H_5NO_3$) | 4-Nitrophenol ($C_6H_5NO_3$) |
| $C_6H_5NO_4$ | 154.0145 | 154.0143 | | 4-Nitrocatechol ($C_6H_5NO_4$) | 4-Nitrocatechol ($C_6H_5NO_4$) |
| $C_7H_7NO_4$ (Iso1[a]) | 168.0302 | 168.0295 | | 2-Methyl-4-nitroresorcinol ($C_7H_7NO_4$) | 2-Methyl-4-nitroresorcinol ($C_7H_7NO_4$) |
| $C_7H_7NO_4$ (Iso2) | 168.0302 | 168.0291 | | 2-Methyl-4-nitroresorcinol ($C_7H_7NO_4$) | 2-Methyl-4-nitroresorcinol ($C_7H_7NO_4$) |
| $C_6H_5NO_5$ | 170.0095 | 170.0087 | | 2-Nitrophloroglucinol ($C_6H_5NO_5$) | 2-Nitrophloroglucinol ($C_6H_5NO_5$) |
| $C_8H_7NO_4$ (Iso1) | 180.0302 | 180.0305 | | 2-Methyl-5-nitrobenzoic acid ($C_8H_7NO_4$) | phenyl cyanate ($C_7H_5NO$) |
| $C_8H_7NO_4$ (Iso2) | 180.0302 | 180.0290 | | 2-Methyl-5-nitrobenzoic acid ($C_8H_7NO_4$) | phenyl cyanate ($C_7H_5NO$) |

[a] Isomer 1; [b] standard compounds used for the quantification of identified N-containing aromatic compounds; [c] standard compounds used to estimate the light absorption of N-containing aromatic compounds.

**Table 2. Continue.**

| Suggested Formula | Theoretical m/z $[M-H]^-$ | Measured m/z $[M-H]^-$ | Proposed structure | Quantified as | Absorbing as |
|---|---|---|---|---|---|
| $C_8H_9NO_4$ (Iso1) | 182.0459 | 182.0467 | | 2-Methyl-4-nitroresorcinol ($C_7H_7NO_4$) | 2-Methyl-4-nitroresorcinol ($C_7H_7NO_4$) |
| $C_8H_9NO_4$ (Iso2) | 182.0459 | 182.0452 | | 2-Methyl-4-nitroresorcinol ($C_7H_7NO_4$) | 2-Methyl-4-nitroresorcinol ($C_7H_7NO_4$) |
| $C_7H_7NO_5$ | 184.0253 | 184.0259 | | 2-Nitrophloroglucinol ($C_6H_5NO_5$) | 2-Nitrophloroglucinol ($C_6H_5NO_5$) |
| $C_{10}H_7NO_3$ | 188.0353 | 188.0356 | | 2-Nitro-1-naphthol ($C_{10}H_7NO_3$) | 2-Nitro-1-naphthol ($C_{10}H_7NO_3$) |
| $C_9H_9NO_4$ (Iso1) | 194.0458 | 194.0461 | | 2,5-Dimethyl-4-nitrobenzoic acid ($C_9H_9NO_4$) | phenyl cyanate ($C_7H_5NO$) |
| $C_9H_9NO_4$ (Iso2) | 194.0458 | 194.0461 | | 2,5-Dimethyl-4-nitrobenzoic acid ($C_9H_9NO_4$) | phenyl cyanate ($C_7H_5NO$) |
| $C_8H_9NO_5$ | 198.0407 | 198.0407 | | 2-Nitrophloroglucinol ($C_6H_5NO_5$) | 2-Nitrophloroglucinol ($C_6H_5NO_5$) |

**Table 2. Continue**

| Suggested Formula | Theoretical m/z [M-H]⁻ | Measured m/z [M-H]⁻ | Proposed structure | Quantified as | Absorbing as |
|---|---|---|---|---|---|
| $C_{10}H_{11}NO_4$ (Iso1) | 208.0615 | 208.0621 |  |  2-Nitrophloroglucinol ($C_6H_5NO_5$) |  phenyl cyanate ($C_7H_5NO$) |
| $C_{10}H_{11}NO_4$ (Iso2) | 208.0615 | 208.0607 |  |  2-Nitrophloroglucinol ($C_6H_5NO_5$) |  phenyl cyanate ($C_7H_5NO$) |
| $C_{10}H_{11}NO_4$ (Iso3) | 208.0615 | 208.0616 |  |  2-Nitrophloroglucinol ($C_6H_5NO_5$) |  2-Nitrophloroglucinol ($C_6H_5NO_5$) |
| $C_{10}H_{11}NO_5$ | 224.0564 | 224.0565 |  |  2-Nitrophloroglucinol ($C_6H_5NO_5$) |  2-Nitrophloroglucinol ($C_6H_5NO_5$) |
| $C_{11}H_{13}NO_5$ | 238.0721 | 238.0722 |  |  2-Nitrophloroglucinol ($C_6H_5NO_5$) |  phenyl cyanate ($C_7H_5NO$) |
| $C_{11}H_{13}NO_6$ | 254.0670 | 254.0670 |  |  2-Nitrophloroglucinol ($C_6H_5NO_5$) |  phenyl cyanate ($C_7H_5NO$) |

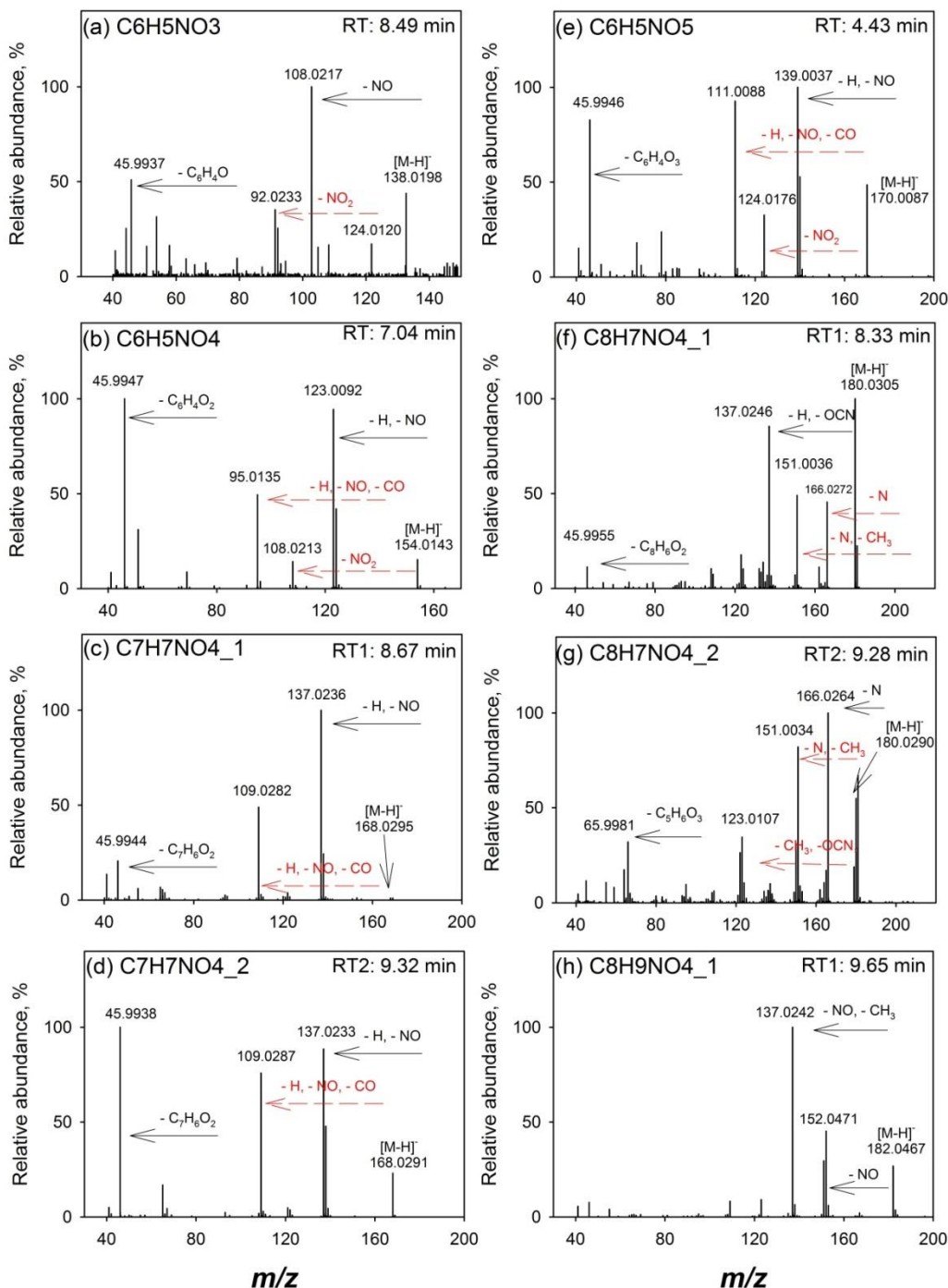

**Figure 1.** Q-ToF MS/MS spectra of (a) $C_6H_5NO_3$, (b) $C_6H_5NO_4$, (c, d) $C_7H_7NO_4$ isomers (e) $C_6H_5NO_5$ , (f, g) $C_8H_7NO_4$ isomers, (h, i) $C_8H_9NO_4$ isomers, (j) $C_7H_7NO_5$, (k) $C_{10}H_7NO_3$, (l, m) $C_9H_9NO_4$ isomers, (n) $C_8H_9NO_5$, (o-q) $C_{10}H_{11}NO_4$ isomers, (r) $C_{10}H_{11}NO_5$, (s) $C_{11}H_{13}NO_5$ and (t) $C_{11}H_{13}NO_6$ identified in the flaming phase sample collected during NC forest 1 experiment, burn 2 (Table S1).

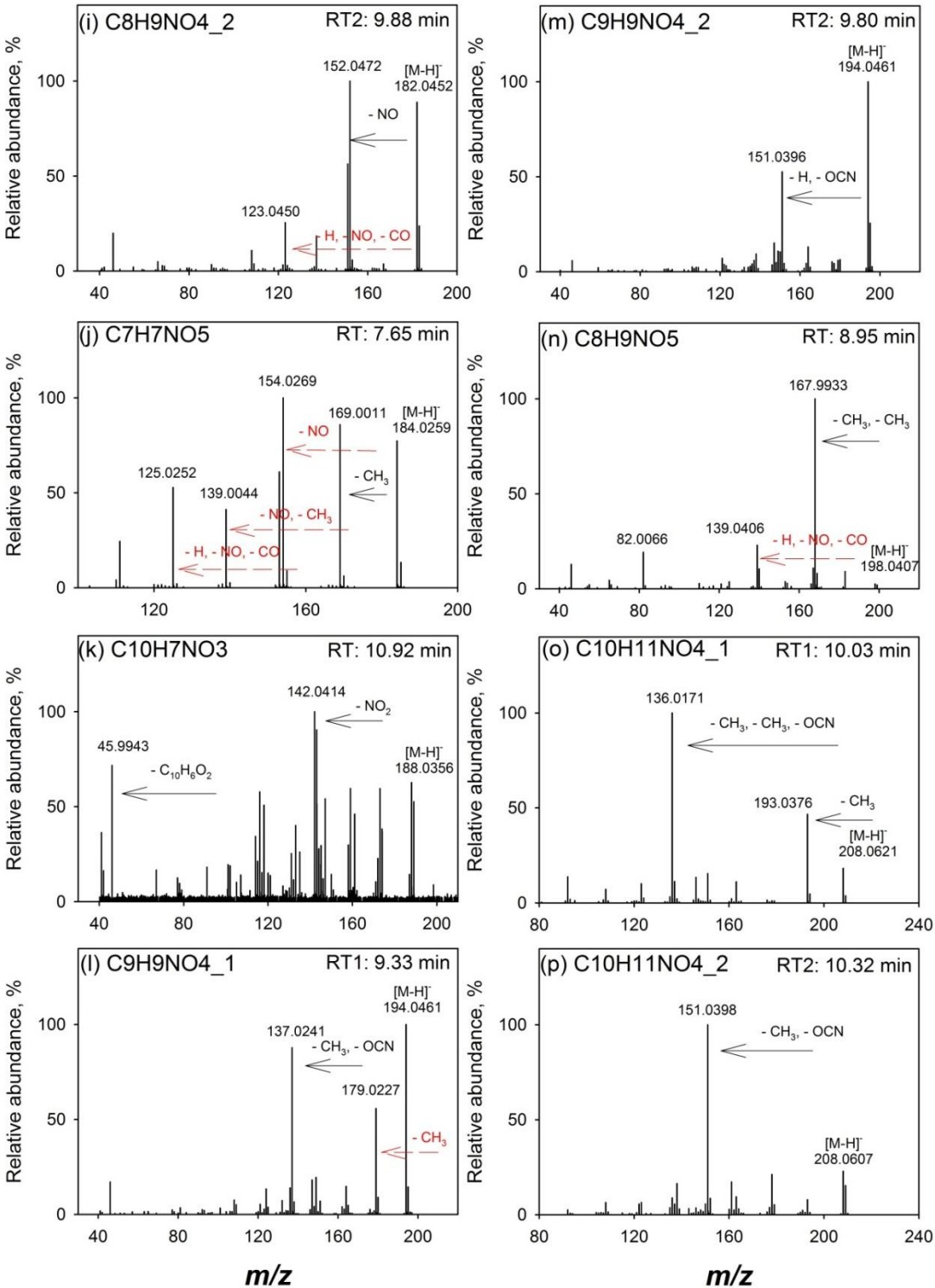

**Figure 1.** Continue

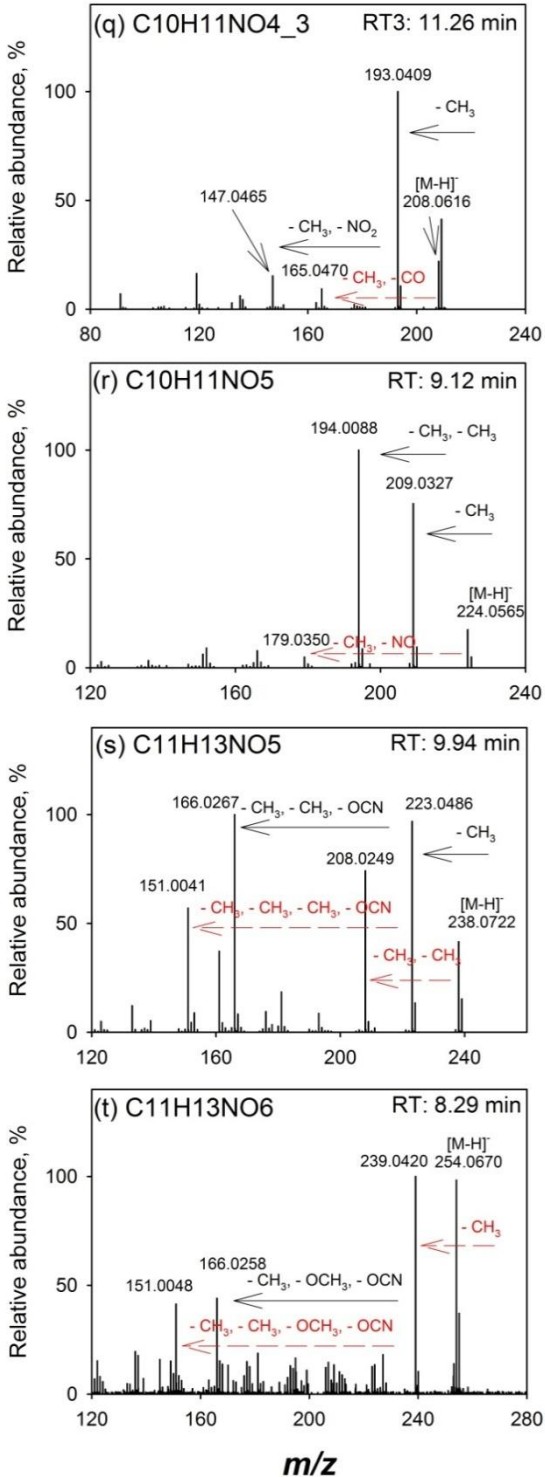

**Figure 1.** Continue

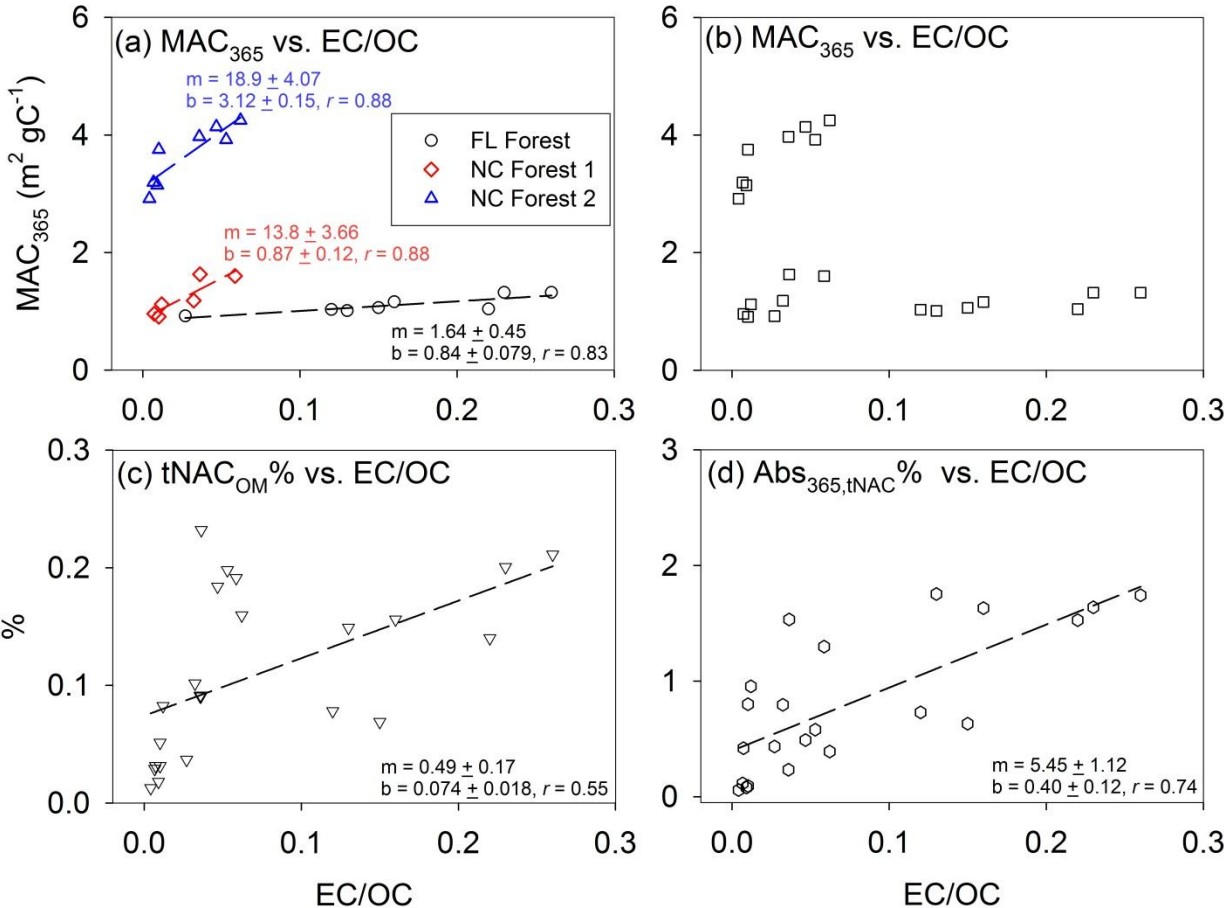

**Figure 2.** Linear regressions of (a) $MAC_{365}$ vs. EC/OC with individual burns data, (b) $MAC_{365}$ vs. EC/OC, (c) $tNAC_{OM}\%$ vs. EC/OC and (d) $Abs_{365,tNAC}\%$ vs. EC/OC with pooled measurements of all the three experiments.

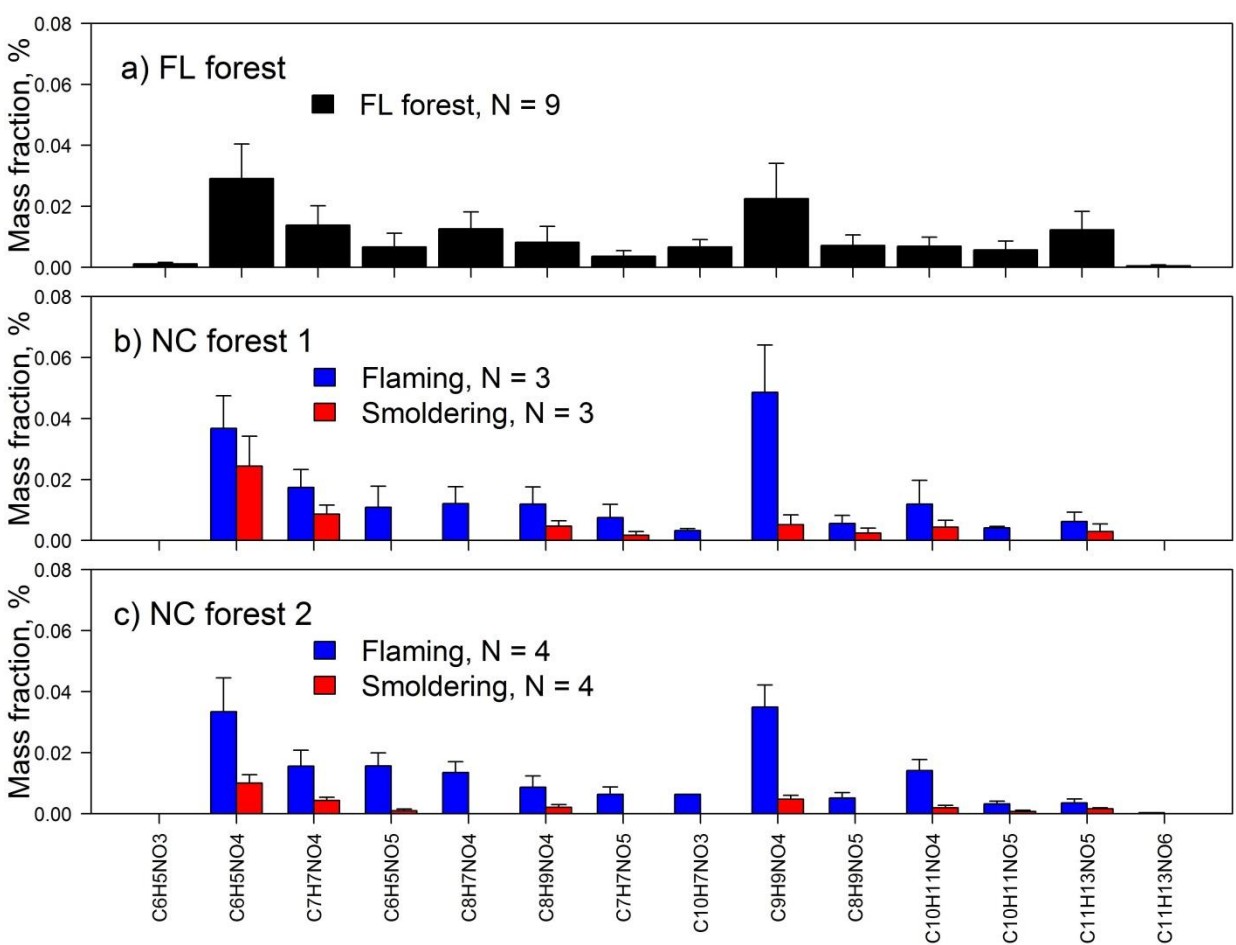

**Figure 3.** Relative mass contributions of identified N-containing aromatic compounds in BB samples collected during (a) FL forest, (b) NC forest 1 and (c) NC forest 2 experiments.

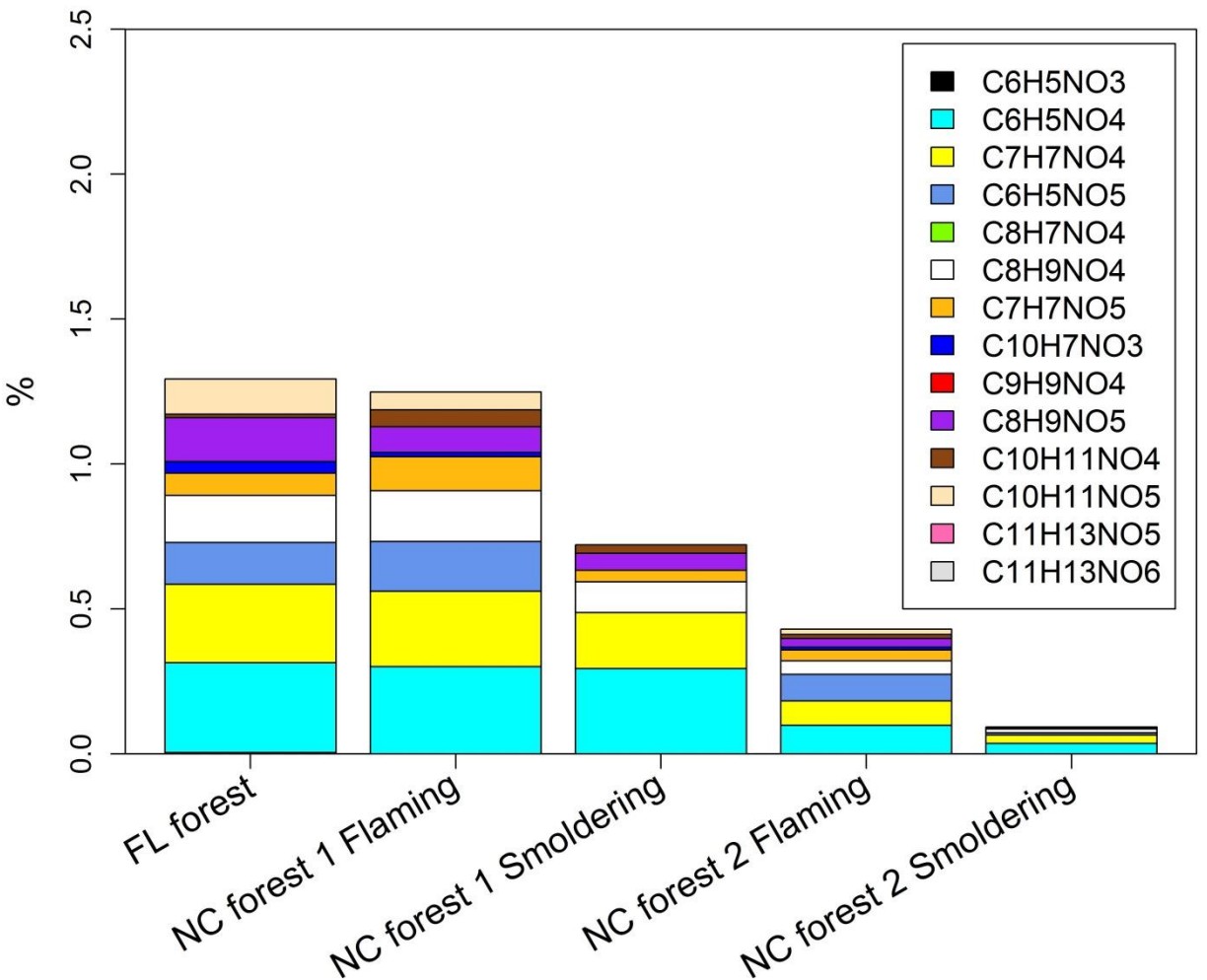

**Figure 4.** Average contributions (%) of N-containing aromatic compounds to $Abs_{365}$ of methanol extractable OC from laboratory biomass burning.