# Peer review of "Composition and light absorption of N-containing aromatic compounds"

_Atmospheric Chemistry and Physics, 2018_

## Referee Comment (RC1) · Anonymous Referee #1 · 8 Nov 2018

Xie et al. 2018 did laboratory burns of two fuel types, and analyzed particles using various offline methods. Most notably, they quantified nitroaromatic compounds (NACs) and their contribution to light absorption using HPLC/DAD-Q-ToFMS and authentic standards. Further characterization was done with MS/MS by comparing fragmentation patterns. The authors also examined the relationship between light absorption and mass of NACs with EC/OC ratio, a proxy for burn conditions. They determined structures for 14 nitrogen-containing chemical formulas and for many of which multiple isomers were observed. Interestingly, four of the nitrogen-containing formulas have only been observed in biomass burning applications before. However, the authors believe that these are not nitroaromatic compounds, but have benzisoxazole structures.

[Figure]

Since these NACs are uniquely observed the authors propose that they could be good tracer compounds.

Overall assessment:

The authors have done careful analytical chemistry in regards to quantifying NACs and confirming structures. They include extraction efficiencies in Table S1, which is very useful for data interpretation. They confirmed structures to the extent possible using MS/MS and surrogate standard compounds. The mass-based contribution of NACs and their contribution to absorbance was very well done with internal standards/surrogates used for quantification. This work is an important contribution to our understanding of biomass burning emissions, and it should definitely be published. I do have some concerns and suggestions for improving the manuscript, as described below.

General Major Comments:

1. The authors should do a little more work to characterize their proposed structures since this is a key conclusion of the paper. They propose benzoxazole structures for the some of the detected compounds based on the analysis of fragmentation patterns and observation of loss of CNO from the ions. However, Giorgi et al. (2004) showed that benzoxazole-based ions lose CO, and to some extent CH3CN, and not CNO during collision induced dissociation. This contradicts the authors' structural assignments. I would recommend buying benzoxizole-based compounds (which are commercially available) and doing MS/MS with them to confirm the fragmentation patterns qualitatively match those observed from BB samples.

2. I would also not so easily dismiss organo-isocyanate structures as is done in the current manuscript. For example, Priestley et al. (2018) observed emissions of methyl isocyanate from biomass burning (p 7697). It is plausible that there may be aromatic isocyanates, and that they would survive extraction. On the other hand, Kaal et al. (2009) found benzoxazole in pyrolyzed charcoal smoke, so the authors can include

this previous observation in the manuscript if there is evidence for the structure.

3. The authors claim these structures are unique to biomass burning and therefore good tracer compounds. However this requires a more comprehensive review of the chamber SOA literature. From a quick search I found that C10H11NO4 was detected in model SOA from the photooxidation of methyl chavicol, an aromatic biogenic precursor, in Pereira et al. (2015). The authors should perhaps do a more thorough job to confirm the uniqueness. Right now this statement that these are unique compounds to biomass burning is weak due to 1) uncertainty in the structures 2) incomplete review of the literature for NACs.

4. I would highly recommend transferring some of the supporting information material to the main manuscript. In particular, the method section in the supporting information should really be in the method section of the main paper. There is no page limitation in ACP, so there is no need to put important information in the SI section.

5. I would also suggest Table S2 be moved, and an example of a CID spectrum for the new structures be added. Right now it is difficult to read the manuscript without referring to the SI. While moving around the figures and tables this can be done strategically to improve the organization of the manuscript. For example, currently Figure 3a is discussed in the first paragraph of the R&D and then discussed again in the last section.

6. It would be helpful to be able to refer to the absorption spectra for these BrC compounds. I would suggest a ðİlJȩmax column in Table S2 or better yet full PDA spectra. This is especially important for those four potential tracer compounds.

7. The authors call these compounds "nitroaromatic compounds", but some of the proposed structures should not be classified as nitroaromatic compounds. They should perhaps name them N-containing aromatic compounds (also abbreviated as NAC)? If so, this should be done throughout the paper.

8. The end of the introduction and conclusion should echo similar messages. The central focus of the paper is not entirely clear. At first, I thought it was to determine the viability of NACs as tracer/marker compounds for biomass burning. At the conclusion, it seems like the main point is to conclude whether fuel type or burn conditions are more important for production of NACs.

Specific Comments:

Pg 4, ln 79-81 It would help if the authors discussed the discrepancy between the reported result (2-18%) and Lin et al. (2017) (50-80%) in the results and discussion section

Pg 5, ln 99-100 This is a weak statement. It would be more appropriate to include a 1-2 sentence summary of the main conclusions of the paper.

Pg 7, ln 165 I suggest moving the sentence on average recoveries of standard compounds to Pg 6, ln 144. It would make more sense there.

Pg 8, ln 178-181 Please consider referencing Fig 3a in the first paragraph. It is much easier to read through with visuals.

Pg 10, ln 230-232 The sentence about quantification with surrogates should be mentioned earlier in the text and in a more systematic way. It is a strength of the paper so it should be better highlighted.

Pg 13, ln 302 Can the authors use nitroaromatic compounds as internal standards to quantify the compounds with benzisoxazole structures? The readers will not really know how these compounds absorb. This could be clarified with including the PDA spectra, as suggested above, and comparing these spectra to the surrogate's spectrum.

Pg 13 Are the identified nitrogen-containing species that are called potentially tracer compounds in this study primarily flaming or smoldering? It was not clear to me based on the writing.

[Figure]

Pg 15 Do burn conditions affect substituents, i.e., the number of OH groups? A brief discussion of this would be useful.

Pg 15, ln 344 It would be helpful to mention here light-absorbing compounds formed at low EC/OC, i.e., tar balls

Figure 3c. I would color code by fuel like in a). Also, there is not a strong correlation. Please remove the trendline.

Technical Corrections:

Pg 2 ln 38 I would I would change "test-specific data" to "individual fires". I think it is less confusing.

Pg 2, ln 48-50 The last sentence of the abstract is unclear to me. Please consider revising.

Pg 3, ln 55-56 Specify OC emissions are specifically OC particle emissions

Pg 3, ln 59 Revise wording of "shorter visible region". Should be near UV, instead of just UV (300-400 nm)

Pg 4, ln 84 Cite Iinuma 2010. It is a critical reference here.

Pg 7, ln 161 Include exactly which internal standards are used.

Pg 12, ln 268 is –> was

Pg 12, ln 271 are –> were

Pg 16, ln 358 Include that these are average tNACOM% by weight

References:

Giorgi, G., Salvini, L. and Ponticelli, F.: Gas phase ion chemistry of the heterocyclic isomers 3-methyl-1,2-benzisoxazole and 2-methyl-1,3-benzoxazole, J. Am. Soc. Mass Spectrom., 15(7), 1005–1013, doi:10.1016/J.JASMS.2004.04.002, 2004.

[Figure]

Iinuma, Y., Böge, O., Gräfe, R. and Herrmann, H.: Methyl-Nitrocatechols: Atmospheric Tracer Compounds for Biomass Burning Secondary Organic Aerosols, Environ. Sci. Technol., 44(22), 8453–8459, doi:10.1021/es102938a, 2010.

Kaal, J., Martínez Cortizas, A. and Nierop, K. G. J.: Characterisation of aged charcoal using a coil probe pyrolysis-GC/MS method optimised for black carbon, J. Anal. Appl. Pyrolysis, 85(1–2), 408–416, doi:10.1016/J.JAAP.2008.11.007, 2009.

Pereira, K. L., Hamilton, J. F., Rickard, A. R., Bloss, W. J., Alam, M. S., Camredon, M., Ward, M. W., Wyche, K. P., Muñoz, A., Vera, T., Vázquez, M., Borrás, E. and Ródenas, M.: Insights into the Formation and Evolution of Individual Compounds in the Particulate Phase during Aromatic Photo-Oxidation, Environ. Sci. Technol., 49(22), 13168–13178, doi:10.1021/acs.est.5b03377, 2015.

Priestley, M., Le Breton, M., Bannan, T. J., Leather, K. E., Bacak, A., Reyes-Villegas, E., De Vocht, F., Shallcross, B. M. A., Brazier, T., Anwar Khan, M., Allan, J., Shallcross, D. E., Coe, H. and Percival, C. J.: Observations of Isocyanate, Amide, Nitrate, and Nitro Compounds From an Anthropogenic Biomass Burning Event Using a ToF-CIMS, J. Geophys. Res. Atmos., 123(14), 7687–7704, doi:10.1002/2017JD027316, 2018.

---

## Referee Comment (RC2) · Anonymous Referee #2 · 15 Nov 2018

This manuscript present a decent study on the light absorption of biomass burning organic aerosols (BBOA) from controlled laboratory burning experiment, with particular focus on the nitroaromatic compounds (NACs), which has been identified as an important light absorbers of OA recently.

Overall this manuscript is well written with clear logic and good English. The only problem on the organization of the content is that the authors keep so many valuable information in the SI, making it impossible to understand their major conclusions without reading the SI. I suggest to move some of them (e.g., Table S2, Figure S2-S4 with some modifications) into the main articles.

[Figure]

Another major issue is the identification of NACs in section 3.2: the authors attempt to get some structural information of NACs through measuring their MS/MS spectra. However, those results should be interpreted more carefully. I don't see any references cited when they discuss the relationship of fragmentation pattern with possible molecular structures. E.g., at line 250-252, "the loss of CNO suggest the skeleton of benzisoxazole……" similar issues can also be found at line 259-261, line 264, line 269-274. The authors may need to find some references investigating the MS/MS spectra of even-electron ions of known standard compounds with similar MS conditions (e.g., https://onlinelibrary.wiley.com/doi/pdf/10.1002/jms.1234) to make educated guess of the structures.

Other problems:

Abstract, line 48-50: this sentence is confusing. I understand that the authors want to say that the burn conditions affect significantly on NACs formation, but slightly on the bulk absorptive properties of BB BrC. However, it reads like the author compare burn condition and bulk adsorptive properties' influence on the formation of NACs.

Section 2.3, HPLC/DAD-MS analysis: the authors use the same analytical method developed in their previous study and described briefly in the current manuscript. It is better to describe the HPLC elution protocols as well so that we don't need to go to another paper if someone want to try the same method or make any comparisons or evaluations about the chromatograph separation.

Line 243-244: the retention time of compound showed in Figure S2b and Figure S3b don't match, with difference $\sim$0.3min, which is too large for a $\sim$20-min length LC chromatogram. If they are the same compound, not only their MS/MS spectra, but also their RTs should also match with each other. Any explanation?

---

## Referee Comment (RC3) · Anonymous Referee #3 · 4 Dec 2018

General comments:

This manuscript presents an interesting study on the presence of nitrogen containing aromatic compounds and their light absorbing properties in laboratory generated biomass burning organic aerosols. I found the manuscript difficult to follow in its current form, mainly because most of important information that supports the authors' discussion is presented in supporting information (Tables S1, S2, and Figure S2). They can be moved to the main manuscript. Apart from the organization of the manuscript, I have three issues that I want the authors to address prior to the acceptance of this manuscript.

[Figure]

Specific major comments:

Table S1 MAC365 for 7/14/2016 sample and lines between 199 and 202
This highlights the difficulties associated with the comparison of the data obtained
from laboratory combustion experiments. As the authors suggest between the lines
199 and 202, the ambient condition appears to be very important for MAC365 values
because the summertime combustion of NC forest 2 shows significantly higher values
for MAC365 than the springtime combustion of NC forest 1. This can only mean
higher absorption coefficients of the NC forest 2 samples than those of the NC forest
1 or lower methanol extractable mass concentrations of the NC forest 2 samples
than those of the NC forest 1. Based on the higher ambient temperature of the NC
forest 2 experiments, I assumed that this originates from the difference in gas/particle
partitioning (i.e. higher gas phase concentrations of MAC365 compounds in NC forest
2 experiment) though it is not too clear to me if this is the case when I see the mass
fractions of MAC365 products depicted in Figure 1 and Table S4. From Figure 2 and
Table S6, it can also deduce that the samples from NC forest 2 combustion contained
highly light absorbing compounds that are not detected in this study. Can the authors
elaborate more in the manuscript? As is now, it is not too clear to me why the MAC365
values are so different when other parameters are relatively similar.

Line 215: Is there a reason for the choice of 1.7 OM/OC factor? In the original paper
of Turpin and Lim (2001), 1.7 was not mentioned as a conversion factor for biomass
burning organic aerosol. Values for fireplace combustion cited in Turpin and Lim (2001)
were between 1.9 and 2.1 that were determined by Schauer (1998). In addition, there
are several more recent values available in the literature.

Identification of the benzisoxazole skeleton
It is not clear to me why the authors attributed the loss of CNO as the presence of the
benzisoxazole skeleton instead of e.g. isocyanates for C9H9NO4 compounds. By los-

ing CNO (actually HCNO) from the benzisoxazole skeleton, one forms highly unstable biradical product ions that aren't likely detected in MS. Can the authors shed light on how the fragments are formed in the revised manuscript?

---

## Author Comment (AC1) · 16 Jan 2019

**Reviewer 1**

Xie et al. 2018 did laboratory burns of two fuel types, and analyzed particles using various offline methods. Most notably, they quantified nitroaromatic compounds (NACs) and their contribution to light absorption using HPLC/DAD-Q-ToFMS and authentic standards. Further characterization was done with MS/MS by comparing fragmentation patterns. The authors also examined the relationship between light absorption and mass of NACs with EC/OC ratio, a proxy for burn conditions. They determined structures for 14 nitrogen-containing chemical formulas and for many of which multiple isomers were observed. Interestingly, four of the nitrogen-containing formulas have only been observed in biomass burning applications before. However, the authors believe that these are not nitroaromatic compounds, but have benzisoxazole structures. Since these NACs are uniquely observed the authors propose that they could be good tracer compounds.

**Overall assessment:**

The authors have done careful analytical chemistry in regards to quantifying NACs and confirming structures. They include extraction efficiencies in Table S1, which is very useful for data interpretation. They confirmed structures to the extent possible using MS/MS and surrogate standard compounds. The mass-based contribution of NACs and their contribution to absorbance was very well done with internal standards/surrogates used for quantification. This work is an important contribution to our understanding of biomass burning emissions, and it should definitely be published. I do have some concerns and suggestions for improving the manuscript, as described below.

*Reply:*

We really appreciate the reviewer's thoughtful comments, which help us to improve the manuscript. We replied to specific comments below.

**General Major Comments:**

**1.** The authors should do a little more work to characterize their proposed structures since this is a key conclusion of the paper. They propose benzoxazole structures for the some of the detected compounds based on the analysis of fragmentation patterns and observation of loss of CNO from the ions. However, Giorgi et al. (2004) showed that benzoxazole-based ions lose CO, and to some extent CH3CN, and not CNO during collision induced dissociation. This contradicts the authors' structural assignments. I would recommend buying benzoxizole-based compounds (which are commercially available) and doing MS/MS with them to confirm the fragmentation patterns qualitatively match those observed from BB samples.

*Reply:*

The reviewer is right. Without enough evidence, we should not attribute the loss of OCN to the existence of benzoxazole structure. According to Giorgi et al. (2004), the MS/MS spectra of 3-methyl-1,2-benzisoxazole and 2-methyl-1,3-benzoxazole at ESI positive ion mode suggested a loss of CO, but not OCN.

[Figure]

**Figure S4.** Mass spectra of (a) phenyl cyanate, (b) benzoxazole, (c) 4-methoxyphenyl isocyanate, and (d) 2,4-dimethoxyphenyl isocyanate with EI mode; MS/MS spectra of (e) 4-methoxyphenyl isocyanate and (f) 2,4-dimethoxyphenyl isocyanate with ESI positive ion mode.

In this work, four standard compounds, including phenyl cyanate ($C_6H_5OCN$), benzoxazole ($C_7H_5NO$), 4-methoxyphenyl isocyanate ($CH_3OC_6H_4NCO$), and 2,4-dimethoxyphenyl isocyanate [$(CH_3O)_2C_6H_3NCO$] were analyzed using a gas chromatography

(Agilent 6890) coupled to a mass spectrometer (Agilent 5975B) under electron ionization (EI, 70 ev) mode. These compounds do not have a phenol structure and cannot be detected using ESI under negative ion mode. The MS/MS spectra of 4-methoxyphenyl isocyanate and 2,4-dimethoxyphenyl isocyanate were obtained by using a modified method (ESI at positive ion mode) for NACs analysis in this work. As shown in Fig. S4a and b, the loss of OCN is observed for phenyl cyanate, but not benzoxazole. In Fig. S4c and d, the ions at $m/z$ 106 and 136 can be produced from the species at $m/z$ 149 and 179 through the loss of $CH_3$ + CO or H + NCO (43 Da). The MS/MS spectra of 4-methoxyphenyl isocyanate and 2,4-dimethoxyphenyl isocyanate (Fig. S4e,f) confirmed the loss of $CH_3$ + CO, and the loss of $CH_3$ reflected the presence of methoxy group. As such, the $C_9H_9NO_4$ compounds identified in this work is expected to contain a phenyl cyanate structure.

These results and discussions were added in the revised manuscript (Pages 14–15, Lines 311–331).

**2.** I would also not so easily dismiss organo-isocyanate structures as is done in the current manuscript. For example, Priestley et al. (2018) observed emissions of methyl isocyanate from biomass burning (p 7697). It is plausible that there may be aromatic isocyanates, and that they would survive extraction. On the other hand, Kaal et al. (2009) found benzoxazole in pyrolyzed charcoal smoke, so the authors can include this previous observation in the manuscript if there is evidence for the structure.

*Reply:*

Yes, correct. As seen in the replies to the reviewer's first comment, we analyzed four additional standard compounds with a structure that might lose OCN using GC-MS and/or LC-MS. In Fig. S4, the loss of OCN group is only observed during the fragmentation of phenyl cyanate. As such, the $C_9H_9NO_4$ compounds identified in this work is expected to contain a phenyl cyanate structure.

**3.** The authors claim these structures are unique to biomass burning and therefore good tracer compounds. However this requires a more comprehensive review of the chamber SOA literature. From a quick search I found that C10H11NO4 was detected in model SOA from the photooxidation of methyl chavicol, an aromatic biogenic precursor, in Pereira et al. (2015). The authors should perhaps do a more thorough job to confirm the uniqueness. Right now this statement that these are unique compounds to biomass burning is weak due to 1) uncertainty in the structures 2) incomplete review of the literature for NACs.

*Reply:*

Yes, $C_{10}H_{11}NO_4$ was observed as 5-methoxy-4-nitro-2-(prop-2-en-1-yl)phenol in SOA from reactions of methyl chavicol and $NO_X$ (Pereira et al. (2015). However, the $C_{10}H_{11}NO_4$ compound observed in this work should have a different structure. Instead of discussing the uniqueness of the four compounds ($C_{10}H_{11}NO_4$, $C_{10}H_{11}NO_5$, $C_{11}H_{13}NO_5$ and $C_{11}H_{13}NO_6$) in BB NACs, we tried to shed lights on the representative functional groups of NACs from BB in the revised manuscript. Before this, we have already mentioned that that the loss of OCN group could be ascribed to the existence of a phenyl cyanate structure.

[Figure]

**Figure S5.** MS/MS spectra of (a) $C_7H_7NO_5$ and (b) $C_8H_9NO_5$ identified from BB in this work and (c, d) the same formula from photo-oxidation of $m$-cresol with $NO_X$.

In the revised manuscript, we stated that "In this work, the NACs formula with molecular weight (MW) < 200 Da (from $C_6H_5NO_3$, 138 Da to $C_8H_9NO_5$, 198 Da) were all identified in secondary organic aerosol (SOA) generated from chamber reactions with $NO_X$ (Xie et al., 2017a). However, the NACs from BB emissions and SOA formations with identical formulas might have different structures. For example, the MS/MS spectra of $C_7H_7NO_5$ and $C_8H_9NO_5$ from BB in this work and aromatic VOCs/$NO_X$ reactions in Xie et al. (2017a) had distinct fragmentation patterns (Fig. S5). In Xie et al. (2017a), the $C_8H_7NO_4$ and $C_9H_9NO_4$ generated from ethylbenzene/$NO_X$ reactions might have fragile structures and their MS/MS spectra were not available. In this work, $C_8H_7NO_4$ and $C_9H_9NO_4$ from BB emissions are more stable and are supposed to have a phenyl

cyanate structure. Among the four NAC formulas with MW > 200 Da identified in this work (Table 2), $C_{10}H_{11}NO_4$ was also observed as 5-methoxy-4-nitro-2-(prop-2-en-1-yl)phenol in SOA from reactions of methyl chavicol and $NO_X$ (Pereira et al. (2015), which cannot be assigned to the $C_{10}H_{11}NO_4$ from BB emissions in this work. Compared to the NACs in aromatic VOCs/$NO_X$ SOA (Iinuma et al., 2010; Lin et al., 2015;Xie et al., 2017a; Pereira et al., 2015), the structures of NACs from BB in this work were characterized by methoxy and cyanate groups. The methoxyphenol structure is a feature in polar organic compounds from BB (Schauer et al., 2001;Simpson et al., 2005;Mazzoleni et al., 2007). The cyanate group was rarely reported in gas- or particle-phase pollutants from BB, which might be a missed feature of BB NACs. Vähä-Savo et al. (2015) found that cyanate could be formed during the thermal conversion (e.g., pyrolysis, gasification) of black liquor, which is the waste product from the kraft process when digesting pulpwood into paper pulp and composed by an aqueous solution of mixed biomass residues. According to Table 2 and Fig. 3, the NACs containing methoxy and/or cyanate groups are predominately generated during the flaming phase in the two NC forest experiments. Before using these compounds as source markers for BB NACs, additional work is warranted to understand their exact structures and lifetimes in the atmosphere. The quantification of these compounds might also be subject to high variability due to the usage of surrogates." (Pages 16–17, Lines 364–390)

**4.** I would highly recommend transferring some of the supporting information material to the main manuscript. In particular, the method section in the supporting information should really be in the method section of the main paper. There is no page limitation in ACP, so there is no need to put important information in the SI section.

*Reply:*

We re-organized the manuscript. All the method details in the original supplementary information was improved and moved into the revised manuscript (Pages 6–7, lines 135–163; pages 8–9, lines 188–194; page 17–18, lines 397–402), including the bulk absorption measurement and calculation, surrogate assignment for NACs quantification, and surrogate assignment for the calculation of NACs contribution to solvent extracts absorption.

**5.** I would also suggest Table S2 be moved, and an example of a CID spectrum for the new structures be added. Right now it is difficult to read the manuscript without referring to the SI. While moving around the figures and tables this can be done strategically to improve the organization of the manuscript. For example, currently Figure 3a is discussed in the first paragraph of the R&D and then discussed again in the last section.

*Reply:*

Yes, Table S2 and Fig. S1 of the original supplementary information were moved into the revised manuscript as Table 2 and Fig. 1. Fig. 3 of the original manuscript is now Fig. 2 in the revised manuscript and first appears in section 3.1, paragraph 2, and right after the discussion on the relationship between light-absorbing properties of BB OC and EC/OC ratios.

In the original manuscripts, Fig. 3a (now Fig. 2a) was not discussed in the first paragraph of section 3.1, and only Table 1 was referred to in the text.

**6.** It would be helpful to be able to refer to the absorption spectra for these BrC compounds. I would suggest a $\lambda_{,}$max column in Table S2 or better yet full PDA spectra. This is especially important for those four potential tracer compounds.

***Reply:***

In this work, a diode array detector (DAD), instead of photodiode array (PDA) spectrophotometry was interfaced with the HPLC and Q-ToFMS. The DAD measurement permits direct identification of chemical compound formulas responsible for light absorption in near UV and visible range (Xie et al., 2017). However, the DAD signals of individual NACs could not be distinguished from bulk absorption of sample extracts (shown in a plot below). This could be due to low contribution of total NACs to the mass (< 1%) and bulk absorption (< 2%) of OM.

[Figure]

**DAD signal of a typical BB sample in this work**

Moreover, the signal peaks in the HPLC/PDA or DAD chromatograms and the corresponding UV-Vis spectra are composed by a mixture of light-absorbing compounds (Lin et al., 2016), some of which are not NACs or even cannot be ionized under ESI positive or negative

ion mode. As such, the $\lambda_{c}$max derived from PDA/DAD spectra cannot reflect the light-absorbing characteristics of individual NACs.

Therefore, the DAD measurements were not exhibited or used for analysis in this work, and a different method applied in Zhang et al. (2013) and Xie et al. (2017) was used to estimate the contribution of individual and total NACs to bulk absorption of extracted OM in BB emissions (Pages 17–18, lines 392–402). The UV-Vis spectra of standard compounds used for the NACs absorption calculation are provided in Fig. S6 of the revised supplementary information.

[Figure]

**Figure S6.** UV/Vis spectra of (a) 4-nitrophenol, 4-nitrocatechol, 2-methyl-4-nitroresorcinol, 2-nitrophloroglucinol, and 2-nitro-1-naphthol at ~1 ng $\mu L^{-1}$ (Xie et al., 2017), and (b) 2,4-dimethoxyphenyl isocyanate, 4-methoxyphenyl isocyanate, and phenyl cyanate at ~10 ng $\mu L^{-1}$.

**7.** The authors call these compounds "nitroaromatic compounds", but some of the proposed structures should not be classified as nitroaromatic compounds. They should perhaps name them N-containing aromatic compounds (also abbreviated as NAC)? If so, this should be done throughout the paper.

*Reply:*

Yes, thanks! The "nitroaromatic compound" was changed into "N-containing aromatic compound" (also abbreviated as NAC) throughout the manuscript.

**8.** The end of the introduction and conclusion should echo similar messages. The central focus of the paper is not entirely clear. At first, I thought it was to determine the viability of NACs as tracer/marker compounds for biomass burning. At the conclusion, it seems like the main point is to conclude whether fuel type or burn conditions are more important for production of NACs.

*Reply:*

Thanks. The central focus of this work was to identify NAC structures specifically related to BB, and quantify the contributions of NACs to BB OM and its solvent extracts abasorption. We revised the conclusion and the last paragraph of the introduction, so as to make them deliver similar information.

The last paragraph of the introduction was changed to:

"The present study attempts to characterize the compositional profile of NACs from BB, identify additional NAC structures in laboratory BB samples, and evaluate the contributions of NACs to bulk absorption of solvent extractable OC from BB. A high-performance liquid chromatograph interfaced to a diode array detector (HPLC/DAD) and quadrupole (Q)-time-of-flight mass spectrometer (ToF-MS) was used to examine NACs in $PM_{2.5}$ (particulate matter with aerodynamic diameter $\leq 2.5$ μm) from three BB experiments. A thermal-optical instrument determined bulk OC and elemental carbon (EC) in the PM, and a UV/Vis spectrometer was used to measure total BrC absorption in methanol extracts of BB $PM_{2.5}$. In this work, a number of NACs formulas with structures that might be specifically related to BB were identified, and the contributions of identified NACs to bulk BrC absorption were calculated. These results shed lights on the light-absorbing characteristics of BB OC at bulk chemical and molecular levels, benefiting the understanding of BrC sources and chromophores." (Pages 4–5, lines 94–105)

The conclusion was changed to:

"The comparisons of light-absorbing properties ($MAC_{365}$, $MAC_{550}$, and $\mathring{A}_{abs}$) of BB OC with EC/OC in this study support that burn conditions are not the only factor impacting BrC absorption. Other factors like fuel type or ambient conditions may also play important roles in

determining BrC absorption from BB. It may be impractical to predict BrC absorption solely based on EC/OC ratios in BB emissions from different fuels or over different seasons. The present study identified fourteen NAC chemical formulas in BB aerosols. The average $tNAC_{OM}$% of the FL forest, NC forest 1 and 2 (flaming and smoldering samples were combined) experiments were $0.13 \pm 0.059$%, $0.13 \pm 0.067$%, and $0.11 \pm 0.017$% by weight, respectively, and the NAC composition was also similar across the three BB experiments. Most of the NACs formulas identified in this work were also observed in simulated SOA generated from chamber reactions of aromatic VOCs with $NO_X$, but the same NAC formula from BB and SOA could not be assigned to the identical compound. In this work, the structures of NACs from BB were characterized by methoxy and cyanate groups, which were predominately generated during the flaming phase and might be an important feature for BB NACs. More work is warranted to understand their exact structures and lifetimes. The average $tNAC_{OM}$% and $Abs_{365,tNAC}$% of the flaming-phase samples were significantly higher ($p < 0.05$) than those of smoldering-phase samples in the two NC forest BB experiments. Unlike the bulk $MAC_{365}$ and $MAC_{550}$, $tNAC_{OM}$% and $Abs_{365,tNAC}$% correlated ($p < 0.05$) with EC/OC for both individual burns and pooled experimental data, suggesting that burn conditions are an important factor in determining NACs formation in BB. Except the compounds with cyanate groups, the NACs identified in this work are likely strong BrC chromophores, as the average contributions of total NACs to bulk $Abs_{365}$ ($0.0087 \pm 0.024$ to $1.22 \pm 0.54$%) are 3–10 times higher than their average mass contributions to OM ($0.023 \pm 0.0089$ to $0.18 \pm 0.067$%). However, more light-absorbing compounds from BB with high MW need to be identified to apportion the unknown fraction ($> 98$%) of BrC absorption." (Pages 19–20, lines 440–464)

**Specific Comments:**

**Comment 1**

Pg 4, ln 79-81 It would help if the authors discussed the discrepancy between the reported result (2-18%) and Lin et al. (2017) (50-80%) in the results and discussion section.

*Reply:*

We added some discussions on the results from Lin et al. (2017) in page 19, lines 423–431.

"Lin et al. (2016, 2017) investigated the light absorption of solvent extractable OC from BB using a combination of HPLC, photodiode array (PDA) spectrophotometry, and high resolution mass spectrometry (HRMS), and attributed a large portion (up to or greater than 50%) of the solvent extracts absorption to a limited number of NACs, of which the MW are mostly lower than 500 Da. However, the signal peaks in the HPLC/PDA chromatograms and the corresponding UV-Vis spectra are likely composed by a mixture of light-absorbing compounds,

some of which are not NACs or even cannot be ionized under ESI positive or negative ion mode. These might lead to an overestimation of NACs contribution to solvent extracts absorption."

Since the results from Lin et al. (2016, 2017) might overestimate the contributions of NACs to solvent extracts of BB OM, we removed the reference in the introduction.

**Comment 2**

Pg 5, ln 99-100 This is a weak statement. It would be more appropriate to include a 1-2 sentence summary of the main conclusions of the paper.

*Reply:*

As we replied to the reviewer's 8[th] general comment, the last paragraph of the introduction was revised. We added a brief summary of the main conclusions and the broad implications of this work in the end.

We stated that "In this work, a number of NACs formulas with structures that might be specifically related to BB were identified, and the contributions of identified NACs to bulk BrC absorption were calculated. These results shed lights on the light-absorbing characteristics of BB OC at bulk chemical and molecular levels, benefiting the understanding of BrC sources and chromophores." (Page 5, lines 101–105)

**Comment 3**

Pg 7, ln 165 I suggest moving the sentence on average recoveries of standard compounds to Pg 6, ln 144. It would make more sense there.

*Reply:*

We re-organized the method section, and moved the method details in the supplementary information to the main text of the manuscript. The field blank and recovery analysis belong to the quality assurance/quality control (QA/QC) of the quantification, and should appear after the method details for NACs quantification. As such, we did not move the description about recoveries of standard compounds.

**Comment 4**

Pg 8, ln 178-181 Please consider referencing Fig 3a in the first paragraph. It is much easier to read through with visuals.

*Reply:*

Fig. 3 of the original manuscript is now Fig. 2 in the revised manuscript and first appears in section 3.1, paragraph 2, and right after the discussion on the relationship between light-absorbing properties of BB OC and EC/OC ratios with previous studies (Page 11, line 240).

In the first paragraph, we mainly compared the light-absorbing properties of OC across the three BB experiments using Table 1. The information presented in Fig. 2 is more related to the second paragraph in section 3.1.

**Comment 5**

Pg 10, ln 230-232 The sentence about quantification with surrogates should be mentioned earlier in the text and in a more systematic way. It is a strength of the paper so it should be better highlighted.

*Reply:*

Yes, we mentioned the use of surrogates with detail in the method section in the revised manuscript (Page 8–9, lines 188–194).

"Due to the lack of authentic standards, most of the NACs in BB samples were quantified using surrogates in this work. In general, the surrogate compound with similar molecular weight (MW) and/or structure was selected for the mass quantification of each identified NAC. Since the standard compound with hydroxyphenyl cyanate structure is not commercially available, $C_8H_7NO_4$ and $C_9H_9NO_4$ were quantified as 2-methyl-5-nitrobenzoic acid ($C_8H_7NO_4$) and 2,5-dimethyl-4-nitrobenzoic acid ($C_9H_9NO_4$), respectively; all the identified NACs with MW > 200 Da were quantified as 2-nitrophloroglucinol ($C_6H_5NO_5$)."

**Comment 6**

Pg 13, ln 302 Can the authors use nitroaromatic compounds as internal standards to quantify the compounds with benzisoxazole structures? The readers will not really know how these compounds absorb. This could be clarified with including the PDA spectra, as suggested above, and comparing these spectra to the surrogate's spectrum.

*Reply:*

As we replied to the reviewer's 1[st] general comments, the identified NACs with a loss of OCN group in the MS/MS spectra were expected to have a phenyl cyanate structure, but not benzoxazole or benzisoxazole structures. According the UV-Vis spectra of 2,4-dimethoxyphenyl isocyanate, 4-methoxyphenyl isocyanate, and phenyl cyanate (Fig. S6b), the contributions of those NACs with cyanate groups to the absorption of bulk solvent extracts were likely to be 0.

As we replied to the reviewer's 6[th] general comments, the signal peaks in the HPLC/PDA or DAD chromatograms and the corresponding UV-Vis spectra are composed by a mixture of light-absorbing compounds (Lin et al., 2016), some of which are not NACs or even cannot be ionized under ESI positive or negative ion mode. As such, the PDA/DAD spectra cannot reflect the light-absorbing characteristics of individual NACs. Fig. S6 provides the UV-Vis spectra of authentic and surrogate standard compounds, which are used to drive the mass absorption coefficients (MAC) of individual NACs in Xie et al. (2017) and this work (NACs with cyanate groups are expected to have a MAC value of 0).

In the revised manuscript, the calculation of NACs contribution to solvent extracts absorption was introduced with more details in pages 17–18, lines 392–402.

"For each sample extract, individual NACs contributions to $Abs_{365}$ ($Abs_{365,NAC}\%$) were calculated using their mass concentrations (ng m$^{-3}$) and the $MAC_{365}$ values of individual compound standards ($MAC_{365,NAC}$), as applied in Zhang et al. (2013) and Xie et al. (2017a). Here, the $MAC_{365,NAC}$ value is OM based with a unit of m$^2$ g$^{-1}$. Each NAC formula was assigned to an authentic or surrogate standard compound to estimate the contribution to $Abs_{365}$ of extracted OM (Table 2). Except the NACs with a phenyl cyanate structure, the standard compounds used for the NACs absorption calculation and mass quantification were the same (Table 2), and their UV-Vis spectra were obtained from Xie et al. (2017a) and shown in Fig. S6a. The UV-Vis spectra of three standard compounds with cyanate or isocyate groups are given in Fig. S6b, and none of them has absorption in the range from 350 to 550 nm. As such, the NACs with cyanate groups identified in this work were supposed to have no contribution to bulk $Abs_{365}$."

**Comment 7**

Pg 13 Are the identified nitrogen-containing species that are called potentially tracer compounds in this study primarily flaming or smoldering? It was not clear to me based on the writing.

*Reply:*

As we replied to the reviewer's 3[rd] general comments, instead of discussing the uniqueness of the four compounds ($C_{10}H_{11}NO_4$, $C_{10}H_{11}NO_5$, $C_{11}H_{13}NO_5$ and $C_{11}H_{13}NO_6$) in BB NACs, we tried to shed lights on the representative functional groups of NACs from BB in the revised manuscript (Page 16–17, lines 364–390). We found that the NACs from BB might be featured by methoxy and cyanate groups. According to Table 2 and Fig. 3 in the revised manuscript, the NACs containing methoxy and/or cyanate groups are predominately generated during the flaming phase of BB experiments.

In the revised manuscript, we mentioned these in section 3.2, page 17, lines 375–385,

"Compared to the NACs in aromatic VOCs/NO$_X$ SOA (Iinuma et al., 2010; Lin et al., 2015;Xie et al., 2017a; Pereira et al., 2015), the structures of NACs from BB in this work were characterized by methoxy and cyanate groups. The methoxyphenol structure is a feature in polar organic compounds from BB (Schauer et al., 2001;Simpson et al., 2005;Mazzoleni et al., 2007). The cyanate group was rarely reported in gas- or particle-phase pollutants from BB, which might be a missed feature of BB NACs. Vähä-Savo et al. (2015) found that cyanate could be formed during the thermal conversion (e.g., pyrolysis, gasification) of black liquor, which is the waste product from the kraft process when digesting pulpwood into paper pulp and composed by an aqueous solution of mixed biomass residues. According to Table 2 and Fig. 3, the NACs containing methoxy and/or cyanate groups are predominately generated during the flaming phase in the two NC forest experiments."

and section 4 (Conclusions), page 20, lines 451–453,

"In this work, the structures of NACs from BB were characterized by methoxy and cyanate groups, which were predominately generated during the flaming phase and might be an important feature for BB NACs."

[Figure]

**Figure 3.** Relative mass contributions of identified N-containing aromatic compounds in BB samples collected during (a) FL forest, (b) NC forest 1 and (c) NC forest 2 experiments.

**Comment 8**

Pg 15 Do burn conditions affect substituents, i.e., the number of OH groups? A brief discussion of this would be useful.

*Reply:*

Yes. According to our replies to the reviewer's previous comments, the identified NACs from BB are featured by methoxy and cyanate groups, which are predominately generated during the flaming phase of the BB experiment. We discussed this in pages 16–17, lines 364–387 and mentioned this in the conclusions (page 20, lines 451–453)

Section 3.4 in the original manuscript was divided and added into previous sections in the revised manuscript separately. Pages 10–11, lines 231–244; page 13, lines 294–298; page 18, lines 415–417.

**Comment 9**

Pg 15, ln 344 It would be helpful to mention here light-absorbing compounds formed at low EC/OC, i.e., tar balls

*Reply:*

Yes. As section 3.4 was divided and added into previous sections, we mentioned the formation of light-absorbing compounds at low EC/OC (e.g., tar balls) with a reference (Chakrabarty et al., 2010) in page 11, lines 243–244.

**Comment 10**

Figure 3c. I would color code by fuel like in a). Also, there is not a strong correlation. Please remove the trend line.

*Reply:*

Fig. 3 in the original manuscript is now Fig. 2 in the revised manuscript. Individual burns data have already been colored by fuel in Fig. S3e of the supplementary information and Fig. 2c here presents the results for pooled experimental data. The correlation in Fig. 2c is not strong but significant ($p < 0.05$). So we kept the trend line in Fig. 2c, and did not color code by fuel in Fig. 2c.

**Technical Corrections:**

**1.** Pg 2 ln 38 I would I would change "test-specific data" to "individual fires". I think it is less confusing.

*Reply:*

The "test-specific data" was changed into "individual burns data" throughout the manuscript.

**2.** Pg 2, ln 48-50 The last sentence of the abstract is unclear to me. Please consider revising.

*Reply:*

The expression was changed into: "The contributions of identified NACs to organic matter (OM) and BrC absorption were significantly higher in flaming-phase samples than those in smoldering-phase samples, and correlated with EC/OC ratio ($p < 0.05$) for both individual burns and pooled experimental data, indicating that the formation of NACs from BB largely depends on burn conditions." (Page 2, lines 47–51)

**3.** Pg 3, ln 55-56 Specify OC emissions are specifically OC particle emissions

*Reply:*

Thanks, we added "particle" in page 3, line 60.

**4.** Pg 3, ln 59 Revise wording of "shorter visible region". Should be near UV, instead of just UV (300-400 nm)

*Reply:*

The original expression was changed into "while light absorption of BB OC increases rapidly from the purple-green region (400–550 nm) to near ultraviolet (UV) region (300–400 nm)." (Page 3, lines 62–64)

**5.** Pg 4, ln 84 Cite Iinuma 2010. It is a critical reference here.

*Reply:*

Thanks, added as suggested in page 4, line 86.

**6.** Pg 7, ln 161 Include exactly which internal standards are used.

*Reply:*

We have already mentioned the compound name and amount of internal standard in page 6, line 143 in the original manuscript, now in page 8, line 167.

**7.** Pg 12, ln 268 is –> was

*Reply:*

Thanks, revised as suggested. Page 15, line 347.

**8.** Pg 12, ln 271 are –> were

*Reply:*

Thanks, revised as suggested. Page 16, line 353.

**9.** Pg 16, ln 358 Include that these are average tNACOM% by weight

*Reply:*

Thanks, revised as suggested. Page 20, line 447.

**References:**

[revised manuscript text omitted]

---

## Author Comment (AC2) · 16 Jan 2019

This manuscript present a decent study on the light absorption of biomass burning organic aerosols (BBOA) from controlled laboratory burning experiment, with particular focus on the nitroaromatic compounds (NACs), which has been identified as an important light absorbers of OA recently.

**1.** Overall this manuscript is well written with clear logic and good English. The only problem on the organization of the content is that the authors keep so many valuable information in the SI, making it impossible to understand their major conclusions without reading the SI. I suggest to move some of them (e.g., Table S2, Figure S2-S4 with some modifications) into the main articles.

*Reply:*

Thanks for the reviewers comments. We re-organized the manuscript, and moved Table S2 and Fig. S2 in the original supplementary information to the main text in the revised manuscript. The method details on bulk absorption measurement and calculation, surrogate assignment for NACs quantification, and surrogate assignment for the calculation of NACs contribution to solvent extracts absorption were also moved to the main text from supplementary information (Pages 6–7, lines 135–163; pages 8–9, lines 188–194; page 17–18, lines 397–402).

**2.** Another major issue is the identification of NACs in section 3.2: the authors attempt to get some structural information of NACs through measuring their MS/MS spectra. However, those results should be interpreted more carefully. I don't see any references cited when they discuss the relationship of fragmentation pattern with possible molecular structures. E.g., at line 250-252, "the loss of CNO suggest the skeleton of benzisoxazole: : :: : :" similar issues can also be found at line 259-261, line 264, line 269-274. The authors may need to find some references investigating the MS/MS spectra of even-electron ions of known standard compounds with similar MS conditions (e.g., https://onlinelibrary.wiley.com/doi/pdf/10.1002/jms.1234) to make educated guess of the structures.

*Reply:*

In the revised manuscript, we added some references and the MS (or MS/MS) spectra of four additional standard compounds to obtain appropriate guess of NACs structures. Finally, the loss of CNO from some NACs was attributed to the existence of a phenyl cyanate structure.

Page 14–15, lines 311–331. "Both MS/MS spectra of the two $C_9H_9NO_4$ isomers reflect the loss of OCN (Fig. 1l,m), suggesting a skeleton of benzoxazole/ benzisoxazole or the existence of a cyanate (–O–C≡N) or isocyanate (–N=C=O) group. Volatile organo-isocyanate structures (e.g., $CH_3NCO$) were identified from anthropogenic biomass burning (Priestley et al., 2018), and benzoxazole structures have been observed in pyrolyzed charcoal smoke (Kaal et al., 2008). Giorgi et al. (2004) investigated the fragmentation of 3-methyl-1,2-benzisoxazole and 2-

methyl-1,3-benzoxazole using a CID technique under different energy frames, and found a loss of CO but not OCN for both of them. In this work, four standard compounds, including phenyl cyanate ($C_6H_5OCN$), benzoxazole ($C_7H_5NO$), 4-methoxyphenyl isocyanate ($CH_3OC_6H_4NCO$), and 2,4-dimethoxyphenyl isocyanate $[(CH_3O)_2C_6H_3NCO]$ were analyzed using a gas chromatography (Agilent 6890) coupled to a mass spectrometer (Agilent 5975B) under electron ionization (EI, 70 ev) mode. These compounds do not have a phenol structure and cannot be detected using ESI under negative ion mode. The MS/MS spectra of 4-methoxyphenyl isocyanate and 2,4-dimethoxyphenyl isocyanate were obtained by using a modified method (ESI at positive ion mode) for NACs analysis in this work. As shown in Fig. S4a and b, the loss of OCN is observed for phenyl cyanate, but not benzoxazole. In Fig. S4c and d, the ions at $m/z$ 106 and 136 can be produced from the species at $m/z$ 149 and 179 through the loss of $CH_3$ + CO or H + NCO (43 Da). The MS/MS spectra of 4-methoxyphenyl isocyanate and 2,4-dimethoxyphenyl isocyanate (Fig. S4e,f) confirmed the loss of $CH_3$ + CO, and the loss of $CH_3$ reflected the presence of methoxy group. As such, the $C_9H_9NO_4$ compounds identified in this work is expected to contain a phenyl cyanate structure."

[Figure]

**Figure S4.** Mass spectra of (a) phenyl cyanate, (b) benzoxazole, (c) 4-methoxyphenyl isocyanate, and (d) 2,4-dimethoxyphenyl isocyanate with EI mode; MS/MS spectra of (e) 4-methoxyphenyl isocyanate and (f) 2,4-dimethoxyphenyl isocyanate with ESI positive ion mode.

Regarding the structure of $C_8H_7NO_4$, the original expression was changed into:

"Like $C_9H_9NO_4$ (Fig. 1l,m), the loss of OCN was observed for the fragmentation of $C_8H_7NO_4$ in the MS/MS spectra (Fig. 1f,g), and a phenyl cyanate structure was proposed (Table 2). However, the fragmentation mechanism associated with the loss of single nitrogen is unknown and warrants further study." (Page 15, 337–340)

The first isomer of $C_8H_9NO_4$ has a dominant ion of $m/z$ 137, reflecting the loss of NO and $CH_3$. Before this, we have mentioned that the loss of $CH_3$ can reflect the presence of methoxy group in pages 14–15, lines 329–330. Referring to MS/MS spectrum of 4-nitrophenol (Fig. S2a), the first $C_8H_9NO_4$ isomer might contain a methyl nitrophenol skeleton with a methoxy group. Page 15, lines 341–344.

The text in lines 269–274 of the original manuscript was changed into:

"In Fig. 1n, the ion at $m/z$ 167 is attributed to the loss of two $CH_3$ from the [M-H]$^-$ ion of $C_8H_9NO_5$, and the loss of H + CO + NO is a common feature shared by several nitrophenol-like compounds (Fig. 1b,c,e,i), so the $C_8H_9NO_5$ compound was identified as dimethoxynitrophenol. The MS/MS spectra of $C_{10}H_{11}NO_4$, $C_{10}H_{11}NO_5$, $C_{11}H_{13}NO_5$, and $C_{11}H_{13}NO_6$ were characterized by the loss of $CH_3$ and/or OCN (Fig. 1o–t), indicting the existence of methoxy and/or cyanate groups (Fig. S4). Although the exact structure of these NACs cannot be determined, their functional groups on the benzene ring were proposed in Table 2 from their fragmentation patterns. " (Pages 15–16, lines 349–356)

In the revised manuscript, the structures of identified NACs in this work were mostly prosed by comparing their MS/MS spectra to authentic and surrogate standard compounds.

**Other problems:**

**1.** Abstract, line 48-50: this sentence is confusing. I understand that the authors want to say that the burn conditions affect significantly on NACs formation, but slightly on the bulk absorptive properties of BB BrC. However, it reads like the author compare burn condition and bulk adsorptive properties' influence on the formation of NACs.

*Reply:*

Thanks for the reviewer's comment. We revised the abstract, and the expression was changed into "The contributions of identified NACs to organic matter (OM) and BrC absorption were significantly higher in flaming-phase samples than those in smoldering-phase samples, and correlated with EC/OC ratio ($p < 0.05$) for both individual burns and pooled experimental data, indicating that the formation of NACs from BB largely depends on burn conditions." (Page 2, lines 47–52)

**2.** Section 2.3, HPLC/DAD-MS analysis: the authors use the same analytical method developed in their previous study and described briefly in the current manuscript. It is better to describe the HPLC elution protocols as well so that we don't need to go to another paper if someone want to try the same method or make any comparisons or evaluations about the chromatograph separation.

*Reply:*

Yes, we added the HPLC elution protocols in page 8, lines 172–175.

"The flow rate of the column was set at 0.2 mL min$^{-1}$, and the gradient separation was conducted with 0.2% acetic acid (v/v) in water (eluent A) and methanol (eluent B). The concentration of eluent B was 25% for the first 3 min, increased to 100% from 3 to 10 min, held at 100% from 10 to 32 min, and then decreased back to 25% from 32 to 37 min."

**3.** Line 243-244: the retention time of compound showed in Figure S2b and Figure S3b don't match, with difference ~0.3min, which is too large for a ~20-min length LC chromatogram. If they are the same compound, not only their MS/MS spectra, but also their RTs should also match with each other. Any explanation?

*Reply:*

Figs. S2 and S3 in the original supplementary information are now Fig.1 in the main text and Fig. S2 in the revised supplementary information, respectively.

As shown in Fig. S2 caption, the MS/MS spectra of standard compounds were obtained from the author's previous study (Xie et al., 2017) on NACs mass and absorption in simulated secondary organic aerosol (SOA). The BB samples in this work and the SOA samples in Xie et al. (2017) were not analyzed together in the same batch. Between these two studies, a few hundred of runs were performed and the properties (e.g., pressure) of guard and/or analytical HPLC columns changed, which is likely the reason for the unmatched retention time between Fig. 1b and Fig. S2b. The retention time of $C_6H_5NO_4$ (nitrocatechol) in Fig. 1b matched with the authentic standard in calibration curve solutions. To clarify that the MS/MS spectrum of 4-nitrocatechol is from a different study, we cited Xie et al. (2017) in page 8, line 186 and page 14, line 306.

**References**

Giorgi, G., Salvini, L., and Ponticelli, F.: Gas phase ion chemistry of the heterocyclic isomers 3-methyl-1,2-benzisoxazole and 2-methyl-1,3-benzoxazole, J. Am. Soc. Mass Spectrom., 15, 1005-1013, https://doi.org/10.1016/j.jasms.2004.04.002, 2004.

Kaal, J., Mart ńez Cortizas, A. and Nierop, K. G. J.: Characterisation of aged charcoal using a coil probe pyrolysis-GC/MS method optimised for black carbon, J. Anal. Appl. Pyrolysis, 85(1–2), 408–416, doi:10.1016/J.JAAP.2008.11.007, 2009.

Priestley, M., Le Breton, M., Bannan, T. J., Leather, K. E., Bacak, A., Reyes-Villegas, E., De Vocht, F., Shallcross, B. M. A., Brazier, T., Anwar Khan, M., Allan, J., Shallcross, D. E., Coe, H., and Percival, C. J.: Observations of isocyanate, amide, nitrate, and nitro compounds from an anthropogenic biomass burning event using a ToF-CIMS, J. Geophys. Res., 123, 7687-7704, doi:10.1002/2017JD027316, 2018.

Xie, M., Chen, X., Hays, M. D., Lewandowski, M., Offenberg, J., Kleindienst, T. E., and Holder, A. L.: Light absorption of secondary organic aerosol: composition and contribution of nitroaromatic compounds, Environ. Sci. Technol., 51, 11607-11616, 10.1021/acs.est.7b03263, 2017.

---

## Author Comment (AC3) · 16 Jan 2019

**General comments:**

This manuscript presents an interesting study on the presence of nitrogen containing aromatic compounds and their light absorbing properties in laboratory generated biomass burning organic aerosols. I found the manuscript difficult to follow in its current form, mainly because most of important information that supports the authors' discussion is presented in supporting information (Tables S1, S2, and Figure S2). They can be moved to the main manuscript. Apart from the organization of the manuscript, I have three issues that I want the authors to address prior to the acceptance of this manuscript.

*Reply:*

Thanks for the reviewer's comments, and we replied to each comment as below.

**Specific major comments:**

**1.** Table S1 MAC365 for 7/14/2016 sample and lines between 199 and 202. This highlights the difficulties associated with the comparison of the data obtained from laboratory combustion experiments. As the authors suggest between the lines 199 and 202, the ambient condition appears to be very important for MAC365 values because the summertime combustion of NC forest 2 shows significantly higher values for MAC365 than the springtime combustion of NC forest 1. This can only mean higher absorption coefficients of the NC forest 2 samples than those of the NC forest 1 or lower methanol extractable mass concentrations of the NC forest 2 samples than those of the NC forest 1. Based on the higher ambient temperature of the NC forest 2 experiments, I assumed that this originates from the difference in gas/particle partitioning (i.e. higher gas phase concentrations of MAC365 compounds in NC forest 2 experiment) though it is not too clear to me if this is the case when I see the mass fractions of MAC365 products depicted in Figure 1 and Table S4. From Figure 2 and Table S6, it can also deduce that the samples from NC forest 2 combustion contained highly light absorbing compounds that are not detected in this study. Can the authors elaborate more in the manuscript? As is now, it is not too clear to me why the MAC365 values are so different when other parameters are relatively similar.

*Reply:*

In the revised manuscript, we added more discussions on the significant difference in bulk $MAC_{365}$ of solvent extractable OC between the two NC forest experiments. We have considered the evaporation of more volatile OC in summer, which might not be the main reason. It is very likely that stronger light-absorbing components are generated from BB in summer than in spring, but the mechanism is unknown and warrants further study.

In pages 11–12, lines 251–263,

"The two NC forest experiments were conducted in spring and summer, respectively, with distinct ambient conditions (Table S1), and their average $MAC_{365}$ values were significantly

($p < 0.05$) different. This could be partly ascribed to the fact that more semi-volatile OC (SVOC) will partition into gas phase in summer with higher ambient temperatures, and the SVOC is less light-absorbing than OC with low volatility (Chen et al., 2010;Saleh et al., 2014). However, if the relative abundance of EC and OC from BB emissions is similar between the two NC forest experiments, the evaporation of SVOCs in summer will lead to higher EC/OC ratios, which is not observed in Table 1. No previous study investigated the seasonal variation in BrC absorption from BB with similar fuel type. Chen et al. (2001) found that the ambient temperature might play a role in EC production from traffic by changing the air density. We suspected that the BB samples from NC forest 2 combustion in summer contained much stronger light-absorbing components than NC forest 1 combustion in spring, although the formation mechanism of these strong BrC components is uncertain and merits further study."

**2.** Line 215: Is there a reason for the choice of 1.7 OM/OC factor? In the original paper of Turpin and Lim (2001), 1.7 was not mentioned as a conversion factor for biomass burning organic aerosol. Values for fireplace combustion cited in Turpin and Lim (2001) were between 1.9 and 2.1 that were determined by Schauer (1998). In addition, there are several more recent values available in the literature.

*Reply:*

Thanks for the reviewer's comment. We used the wrong reference in the original manuscript, and we replaced Tupin and Lim (2001) with Reff et al. (2009). In the supplementary information of Reff et al. (2009), they went through a summary of the OM/OC ratio of various sources and they arrive at a value of 1.7 for biomass burning/wood burning.

**3.** Identification of the benzisoxazole skeleton It is not clear to me why the authors attributed the loss of CNO as the presence of the benzisoxazole skeleton instead of e.g. isocyanates for C9H9NO4 compounds. By losing CNO (actually HCNO) from the benzisoxazole skeleton, one forms highly unstable biradical product ions that aren't likely detected in MS. Can the authors shed light on how the fragments are formed in the revised manuscript?

*Reply:*

Yes, we should not attribute the loss of OCN to a benzoxazole structure without enough evidence. In the revised work, four standard compounds, including phenyl cyanate ($C_6H_5OCN$), benzoxazole ($C_7H_5NO$), 4-methoxyphenyl isocyanate ($CH_3OC_6H_4NCO$) and 2,4-dimethoxyphenyl isocyanate [$(CH_3O)_2C_6H_3NCO$], were analyzed using a gas chromatography (Agilent 6890) coupled to a mass spectrometer (Agilent 5975B) under electron ionization (EI, 70 ev) mode. These compounds do not have a phenol structure and cannot be detected using ESI

under negative ion mode. The MS/MS spectra of 4-methoxyphenyl isocyanate and 2,4-dimethoxyphenyl isocyanate were obtained by using a modified method (ESI at positive ion mode) for NACs analysis in this work. As shown in Fig. S4a and b in the revised supplementary information (See below), the loss of OCN is observed for phenyl cyanate, but not benzoxazole. In Fig. S4c and d, the ions at $m/z$ 106 and 136 can be produced from the species at $m/z$ 149 and 179 through the loss of $CH_3$ + CO or H + NCO (43 Da). The MS/MS spectra of 4-methoxyphenyl isocyanate and 2,4-dimethoxyphenyl isocyanate (Fig. S4e,f) confirmed the loss of $CH_3$ + CO, and the loss of $CH_3$ reflected the presence of methoxy group. As such, the $C_9H_9NO_4$ compounds identified in this work is expected to contain a phenyl cyanate structure.

These results and discussions were added in the revised manuscript (Pages 14–15, lines 311–331).

[Figure]

**Figure S4.** Mass spectra of (a) phenyl cyanate, (b) benzoxazole, (c) 4-methoxyphenyl isocyanate, and (d) 2,4-dimethoxyphenyl isocyanate with EI mode; MS/MS spectra of (e) 4-methoxyphenyl isocyanate and (f) 2,4-dimethoxyphenyl isocyanate with ESI positive ion mode.

**References**

Chen, L., W.A, Doddridge, B. G., Dickerson, R. R., Chow, J. C., Mueller, P. K., Quinn, J., and Butler, W. A.: Seasonal variations in elemental carbon aerosol, carbon monoxide and sulfur dioxide: Implications for sources, Geophys. Res. Lett., 28, 1711-1714, 10.1029/2000gl012354, 2001.

Chen, Y., and Bond, T. C.: Light absorption by organic carbon from wood combustion, Atmos. Chem. Phys., 10, 1773-1787, 10.5194/acp-10-1773-2010, 2010.

Reff, A., Bhave, P. V., Simon, H., Pace, T. G., Pouliot, G. A., Mobley, J. D., and Houyoux, M.: Emissions inventory of PM2.5 trace elements across the United States, Environ. Sci. Technol., 43, 5790-5796, 10.1021/es802930x, 2009.

Saleh, R., Robinson, E. S., Tkacik, D. S., Ahern, A. T., Liu, S., Aiken, A. C., Sullivan, R. C., Presto, A. A., Dubey, M. K., Yokelson, R. J., Donahue, N. M., and Robinson, A. L.: Brownness of organics in aerosols from biomass burning linked to their black carbon content, Nature Geosci., 7, 647-650, 10.1038/ngeo2220, 2014.

Turpin, B. J., and Lim, H.-J.: Species contributions to $PM_{2.5}$ mass concentrations: revisiting common assumptions for estimating organic mass, Aerosol Sci. Tech., 35, 602-610, 10.1080/02786820119445, 2001.

---

## Referee Report (RR1)

Thank you for your response. The paper is more cohesive and focused.

Specific comments:

Pg 3, lines 58-68 Only absorption by particulate OC (brown carbon) is discussed in the introduction. This can be misleading when discussing OC particle emissions from BB burning. According to Selimovic et al. 2018, for all fuel types, scattering is the dominant effect relative to absorption at 401 nm, which is assumed to be entirely organics (see Table 3). Scattering also affects the radiation balance of the Earth, and while it is not discussed in the paper, it should be introduced to help put your work into context.

Pg 18, lines 411-412 I would not characterize the cyanate compounds as highly absorbing. Please clarify that nitroaromatics are highly absorbing (even more so than the calculation, since cyanates are lumped in with them if I understand correctly.)

Figure S6. The absorbance for phenyl cyanate appears to be negative at certain wavelengths. This doesn't make physical sense. Please provide an explanation or consider revising.

Technical comments:

Pg 2 Lines 38-40 Please change sentence to "However, the pooled experimental data indicated that EC/OC alone cannot explain the BB BrC absorption." I think this is clearer.

Pg 3, Line 65 Delete "caused"

Pg 5, lines 104-105 should be "shed light"

Pg 19, lines 423-431 Please consider revising for a more concise explanation. What I read is that in Lin et al. 2016, 2017 uses absorbance at a particular retention time (implicating possible interferences or coelution) to calculate the absorbance fraction, whereas this paper uses standards or surrogates to calculate absorption for each molecule. These different approaches gave different results.

---

## Author Response (AR2)

**Specific comments:**

**1.** Pg 3, lines 58-68 Only absorption by particulate OC (brown carbon) is discussed in the introduction. This can be misleading when discussing OC particle emissions from BB burning. According to Selimovic et al. 2018, for all fuel types, scattering is the dominant effect relative to absorption at 401 nm, which is assumed to be entirely organics (see Table 3). Scattering also affects the radiation balance of the Earth, and while it is not discussed in the paper, it should be introduced to help put your work into context.

**Reply:**

Both the light scattering and absorption of organic carbon (OC) play important roles in the Earth's radiative balance (Ramanathan et al., 2001;Anderson et al., 2003;Bond and Bergstrom, 2006). Selimovic et al. (2018) found that the BrC absorption is responsible for at least half and up to two-thirds of the absorption at 401 nm, and inferred that the net radiative forcing of biomass burning (BB) is not cooling or neutral, but warming if BrC is long-lived enough. The Table 3 in Selimovic et al. (2018) only provides single-scattering albedo (SSA) and Ångström absorption exponent (AAE) of BB smoke, from which we cannot conclude that scattering is the dominant effect relative to absorption at 401 nm.

In the revised manuscript, we added some text and revised a few words on OC scattering in the introduction. (Page 3, lines 62–69)

"OC in particulate matter (PM) is commonly treated as purely light scattering component in global climate models (Chung and Seinfeld, 2002;Myhre et al., 2013). Recent field and laboratory studies found that the light absorption of BB OC increases rapidly from the purplegreen region (400–550 nm) to near ultraviolet (UV) region (300–400 nm) (Kirchstetter et al., 2004;Laskin et al., 2015;Chakrabarty et al., 2016;Xie et al., 2017b). The light absorption and scattering caused by BC and OC from BB can directly affect the Earth's radiative balance (Ramanathan et al., 2001;Anderson et al., 2003;Bond and Bergstrom, 2006),......"

**2.** Pg 18, lines 411-412 I would not characterize the cyanate compounds as highly absorbing. Please clarify that nitroaromatics are highly absorbing (even more so than the calculation, since cyanates are lumped in with them if I understand correctly.)

**Reply:**

Yes, the cyanate compounds were expected to have no contribution to  $Abs_{365}$  of sample extracts. That's why we stated: "....indicating that the identified NACs with contributions to  $Abs_{365}$  are strong BrC chromophores." (Page 18, lines 414–415) We emphasized that the NACs with contributions to  $Abs_{365}$  are highly absorbing.

To make it clearer, we added "not including those with cyanate groups" in page 18, line 415.

**3.** Figure S6. The absorbance for phenyl cyanate appears to be negative at certain wavelengths. This doesn't make physical sense. Please provide an explanation or consider revising.

Reply:

Yes, the negative absorbance makes no physical sense, which might be resulted from instrument error. After calibrating the instrument and cleaning the cuvette thoroughly, we measured the light absorption of 2,4-dimethoxyphenyl isocyanate, 4-methoxyphenyl isocyanate, and phenyl cyanate in methanol at ~10 ng  $\mu$ L-1 again. Figure S6b was revised and none of the three compounds had negative absorption in the wavelength range. Their absorption were all 0 from 350 to 550 nm.

**Figure S6.** UV/Vis spectra of (a) 4-nitrophenol, 4-nitrocatechol, 2-methyl-4-nitroresorcinol, 2-nitrophloroglucinol, and 2-nitro-1-naphthol at ~1 ng  $\mu L^{-1}$ , and (b) 2,4-dimethoxyphenyl isocyanate, 4-methoxyphenyl isocyanate, and phenyl cyanate at ~10 ng  $\mu L^{-1}$ .

**Technical comments:**

**1.** Pg 2 Lines 38-40 Please change sentence to "However, the pooled experimental data indicated that EC/OC alone cannot explain the BB BrC absorption." I think this is clearer.

**Reply:**

Yes, thanks! The sentence was changed as suggested.

"However, the pooled experimental data indicated that EC/OC alone cannot explain the BB BrC absorption." (Page 2, lines 38–39)

2. Pg 3, Line 65 Delete "caused"

**Reply:**

It was deleted as suggested.

**3.** Pg 5, lines 104-105 should be "shed light"

**Reply:**

Thanks! It was revised as suggested (Page 5, line 107).

**4.** Pg 19, lines 423-431 Please consider revising for a more concise explanation. What I read is that in Lin et al. 2016, 2017 uses absorbance at a particular retention time (implicating possible interferences or coelution) to calculate the absorbance fraction, whereas this paper uses standards or surrogates to calculate absorption for each molecule. These different approaches gave different results.

**Reply:**

In Lin et al. (2016, 2017), the absorbance at a particular retention time is contributed by a mixture of light-absorbing compounds due to coelution. As we mentioned in the text, these compounds are not all NACs, and some of them might not be detectable using HPLC-ESI-MS. So Lin et al. (2016, 2017) should overestimate the contribution of NACs to solvent extracts absorption for BB aerosol.

Compared to this study, Lin et al. (2016, 2017) used different instruments and analytical methods to estimate the contribution of NACs to solvent extracts absorption, which is more than 10 times higher than this work and might be biased due to coelution of light-absorbing compounds. That's why we discussed such big discrepancy with details.

In the revised manuscript, these discussions were changed into:

"Lin et al. (2016, 2017) calculated the absorbance fraction contributed by NACs in BB OC based on signal peaks at particular retention times in HPLC/photodiode array (PDA) spectrophotometry chromatograms, and attributed a large portion (up to or greater than 50%) of the solvent extracts absorption to a limited number of NACs with MW mostly lower than 500 Da. However, the absorbance signals in HPLC/PDA chromatograms are composed by a mixture of light-absorbing compounds due to coelution, and some of them are not NACs or even cannot be ionized with ESI. In this study, standards or surrogates were used to calculate absorption for individual NACs molecules. These different approaches gave different results." (Page 19, lines 426–434.

**References**

- Anderson, T. L., Charlson, R. J., Schwartz, S. E., Knutti, R., Boucher, O., Rodhe, H., and Heintzenberg, J.: Climate forcing by aerosols--a hazy picture, Science, 300, 1103-1104, 10.1126/science.1084777, 2003.
- Bond, T. C., and Bergstrom, R. W.: Light absorption by carbonaceous particles: An investigative review, Aerosol Sci. Technol., 40, 27-67, 10.1080/02786820500421521, 2006.
- Ramanathan, V., Crutzen, P. J., Kiehl, J. T., and Rosenfeld, D.: Aerosols, climate, and the hydrological cycle, Science, 294, 2119-2124, 10.1126/science.1064034, 2001.
- Selimovic, V., Yokelson, R. J., Warneke, C., Roberts, J. M., de Gouw, J., Reardon, J., and Griffith, D. W. T.: Aerosol optical properties and trace gas emissions by PAX and OP-FTIR for laboratory-simulated western US wildfires during FIREX, Atmos. Chem. Phys., 18, 2929-2948, 10.5194/acp-18-2929-2018, 2018.

[revised manuscript text omitted]
_3$                                         | 138.0196                 | 138.0198              | OH
O'N'=O                                         | 4-Nitrophenol (C 6 H₅NO 3 )                      | 4-Nitrophenol (C 6 H₅NO 3 )                      |
| $C_6H_5NO_4$                                         | 154.0145                 | 154.0143              | OH
OH
O-M SO                                   | 4-Nitrocatechol (C 6 H 5 NO 4 )       | 4-Nitrocatechol (C 6 H 5 NO 4 )       |
| C7H7NO4 (Iso1 ª )                         | 168.0302                 | 168.0295              | HO CH 3
o -N × o | 2-Methyl-4-nitroresorcinol                                             | 2-Methyl-4-nitroresorcinol                                             |
| C 7 H 7 NO 4 (Iso2) | 168.0302                 | 168.0291              | OFN CH                                               | 2-Methyl-4-nitroresorcinol
(C 7 H 5 )         | 2-Methyl-4-nitroresorcinol                                             |
| $C_6H_5NO_5$                                         | 170.0095                 | 170.0087              | HO
HO
O
N
O                              | 2-Nitrophloroglucinol (C 6 H 5 NO 5 ) | 2-Nitrophloroglucinol (C 6 H 5 NO 5 ) |
| $C_8H_7NO_4$ (Iso1)                                  | 180.0302                 | 180.0305              |                                                      | 2-Methyl-5-nitrobenzoic acid                                           | phenyl cyanate
(C7H₅NO)                                             |
| C 8 H 7 NO 4 (Iso2) | 180.0302                 | 180.0290              | HO HO N
I CH 3                         | 2-Methyl-5-nitrobenzoic acid                                           | phenyl cyanate
(C 7 H₅NO)                                |

Table 2. Identified N-containing aromatic compounds by HPLC/ESI-Q-ToFMS from laboratory biomass burning in this study.

a Isomer 1; b standard compounds used for the quantification of identified N-containing aromatic compounds; c standard compounds used to estimate the light absorption of N-containing aromatic compounds.

**Table 2. Continue.**

| Suggested Formula                                    | Theoretical m/z
[M-H] - | Measured m/z
[M-H] - | Proposed structure                                                                                           | Quantified as                                                                        | Absorbing as                                                                   |
|------------------------------------------------------|---------------------------------------|------------------------------------|--------------------------------------------------------------------------------------------------------------|--------------------------------------------------------------------------------------|--------------------------------------------------------------------------------|
| $C_8H_9NO_4$ (Iso1)                                  | 182.0459                              | 182.0467                           | H 3 C H3 O H
H 3 C - N +,2 O | 2-Methyl-4-nitroresorcinol
(C 7 H 7 NO 4 )       | 2-Methyl-4-nitroresorcinol
(C 7 H 7 NO 4 ) |
| C 8 H 9 NO 4 (Iso2) | 182.0459                              | 182.0452                           | CH3 OH
O-N * O                                                                                 | 2-Methyl-4-nitroresorcinol
(C 7 H 7 NO 4 )       | 2-Methyl-4-nitroresorcinol
(C 7 H 7 NO 4 ) |
| C7H7NO5                                              | 184.0253                              | 184.0259                           | CH 3 OH
O
ON
O                                                                           | 2-Nitrophloroglucinol (C 6 H 5 NO 5 )               | 2-Nitrophloroglucinol (C6H5NO5)                                                |
| $C_{10}H_7NO_3$                                      | 188.0353                              | 188.0356                           | N O                                                                                               | 2-Nitro-1-naphthol (C 10 H 7 NO 3 )                 | 2-Nitro-1-naphthol (C 10 H 7 NO 3 )           |
| $C_9H_9NO_4$ (Iso1)                                  | 194.0458                              | 194.0461                           | H 3 C O
H 3 C O                                               | 2,5-Dimethyl-4-nitrobenzoic acid
(C 9 H 9 NO 4 ) | phenyl cyanate
(C7H₅NO)                                                     |
| C₀H₀NO₄ (Iso2)                                       | 194.0458                              | 194.0461                           |                                                                                                              | 2,5-Dimethyl-4-nitrobenzoic acid                                                     | phenyl cyanate
(C7H₅NO)                                                     |
| C 8 H 9 NO 5        | 198.0407                              | 198.0407                           | H 3 C 0 OH                                                                             | 2-Nitrophloroglucinol (C 6 H 5 NO 5 )               | $\underbrace{\overset{OH}{\underset{HO}{}{}{}{}{}{}{}{$                        |

**Table 2. Continue**

| Suggested Formula                                      | Theoretical m/z
[M-H] - | Measured m/z
[M-H] - | Proposed structure                                                                                            | Quantified as                                                                                                                                                                                                                                                                                                                                                                                                                                                                                                                                                                                                                                                                                                                                                                                                                                                                                                                                                                                                                                                                                                                                                                                                                                                                                                                                                                                                                                                                                                                                                                                                                                                                                                                                                                                                                                                                                                                                                                                                                                                                                                                                                                                                                                                                                                                                                                                                                                                                                                                                                                                                                                                                                              | Absorbing as                            |
|--------------------------------------------------------|---------------------------------------|------------------------------------|---------------------------------------------------------------------------------------------------------------|------------------------------------------------------------------------------------------------------------------------------------------------------------------------------------------------------------------------------------------------------------------------------------------------------------------------------------------------------------------------------------------------------------------------------------------------------------------------------------------------------------------------------------------------------------------------------------------------------------------------------------------------------------------------------------------------------------------------------------------------------------------------------------------------------------------------------------------------------------------------------------------------------------------------------------------------------------------------------------------------------------------------------------------------------------------------------------------------------------------------------------------------------------------------------------------------------------------------------------------------------------------------------------------------------------------------------------------------------------------------------------------------------------------------------------------------------------------------------------------------------------------------------------------------------------------------------------------------------------------------------------------------------------------------------------------------------------------------------------------------------------------------------------------------------------------------------------------------------------------------------------------------------------------------------------------------------------------------------------------------------------------------------------------------------------------------------------------------------------------------------------------------------------------------------------------------------------------------------------------------------------------------------------------------------------------------------------------------------------------------------------------------------------------------------------------------------------------------------------------------------------------------------------------------------------------------------------------------------------------------------------------------------------------------------------------------------------|-----------------------------------------|
| C 10 H 11 NO 4 (Iso1) | 208.0615                              | 208.0621                           | CH3 OH
H3C N                                                                                               | $\begin{array}{c} \overset{\text{off}}{\underset{H_0}{\overset{H_1}{\overset{H_2}{\overset{H_3}{\overset{H_4}{\overset{H_5}{\overset{H_5}{\overset{H_5}{\overset{H_5}{\overset{H_5}{\overset{H_5}{\overset{H_5}{\overset{H_5}{\overset{H_5}{\overset{H_5}{\overset{H_5}{\overset{H_5}{\overset{H_5}{\overset{H_5}{\overset{H_5}{\overset{H_5}{\overset{H_5}{\overset{H_5}{\overset{H_5}{\overset{H_5}{\overset{H_5}{\overset{H_5}{\overset{H_5}{\overset{H_5}{\overset{H_5}{\overset{H_5}{\overset{H_5}{\overset{H_5}{\overset{H_5}{\overset{H_5}{\overset{H_5}{\overset{H_5}{\overset{H_5}{\overset{H_5}{\overset{H_5}{\overset{H_5}{\overset{H_5}{\overset{H_5}{\overset{H_5}{\overset{H_5}{\overset{H_5}{\overset{H_5}{\overset{H_5}{\overset{H_5}{\overset{H_5}{\overset{H_5}{\overset{H_5}{\overset{H_5}{\overset{H_5}{\overset{H_5}{\overset{H_5}{\overset{H_5}{\overset{H_5}{\overset{H_5}{\overset{H_5}{\overset{H_5}{\overset{H_5}{\overset{H_5}{\overset{H_5}{\overset{H_5}{\overset{H_5}{\overset{H_5}{\overset{H_5}{\overset{H_5}{\overset{H_5}{\overset{H_5}{\overset{H_5}{\overset{H_5}{\overset{H_5}{\overset{H_5}{\overset{H_5}{\overset{H_5}{\overset{H_5}{\overset{H_5}{\overset{H_5}{\overset{H_5}{\overset{H_5}{\overset{H_5}{\overset{H_5}{\overset{H_5}{\overset{H_5}{\overset{H_5}{\overset{H_5}{\overset{H_5}{\overset{H_5}{\overset{H_5}{\overset{H_5}{\overset{H_5}{\overset{H_5}{\overset{H_5}{\overset{H_5}{\overset{H_5}{\overset{H_5}{\overset{H_5}{\overset{H_5}{\overset{H_5}{\overset{H_5}{\overset{H_5}{\overset{H_5}{\overset{H_5}{\overset{H_5}{\overset{H_5}{\overset{H_5}{\overset{H_5}{\overset{H_5}{\overset{H_5}{\overset{H_5}{\overset{H_5}{\overset{H_5}{\overset{H_5}{\overset{H_5}{\overset{H_5}{\overset{H_5}{\overset{H_5}{\overset{H_5}{\overset{H_5}{\overset{H_5}{\overset{H_5}{\overset{H_5}{\overset{H_5}{\overset{H_5}{\overset{H_5}{\overset{H_5}{\overset{H_5}{\overset{H_5}{\overset{H_5}{\overset{H_5}{\overset{H_5}{\overset{H_5}{\overset{H_5}{\overset{H_5}{\overset{H_5}{\overset{H_5}{\overset{H_5}{\overset{H_5}{\overset{H_5}{\overset{H_5}{\overset{H_5}{\overset{H_5}{\overset{H_5}{\overset{H_5}{\overset{H_5}{\overset{H_5}{\overset{H_5}{\overset{H_5}{\overset{H_5}{\overset{H_5}{\overset{H_5}{\overset{H_5}{\overset{H_5}{\overset{H_5}{\overset{H_5}{\overset{H_5}{\overset{H_5}{\overset{H_5}{\overset{H_5}{\overset{H_5}{\overset{H_5}{\overset{H_5}{\overset{H_5}{\overset{H_5}{\overset{H_5}{\overset{H_5}{\overset{H_5}{\overset{H_5}{\overset{H_5}{\overset{H_5}{\overset{H_5}}{\overset{H_5}}{\overset{H_5}}{\overset{H_5}}{\overset{H_5}}{\overset{H_5}}{\overset{H_5}}{\overset{H_5}}}}}}}}}}}}}}}}}}}}}}}}}}}}}}}}}}}}$ | phenyl cyanate
(C7H₅NO)              |
| C 10 H 11 NO 4 (Iso2) | 208.0615                              | 208.0607                           | Ho
Ho
H 2 C                                                                                  | $\begin{array}{c} \overset{\text{OH}}{\underset{HO}{}{}{}{}{}{}{}{$                                                                                                                                                                                                                                                                                                                                                                                                                                                                                                                                                                                                                                                                                                                                                                                                                                                                                                                                                                                                                                                                                                                                                                                                                                                                                                                                                                                                                                                                                                                                                                                                                                                                                                                                                                                                                                                                                                                                                                                                                                                                                                                                                                                                                                                                                                                                                                                                                                                                                                                                                                                                                                        | phenyl cyanate
(C7HsNO)              |
| C 10 H 11 NO 4 (Iso3) | 208.0615                              | 208.0616                           | о
н 3 с сн 3                                                                         | $\begin{array}{c} \overset{\overset{\overset{\overset{\overset{\overset{\overset{\overset{}}}}}}}}{}}{}_{H_0}}}{}\\ 2\text{-Nitrophloroglucinol}}\\ (C_6H_5NO_5)\end{array}$                                                                                                                                                                                                                                                                                                                                                                                                                                                                                                                                                                                                                                                                                                                                                                                                                                                                                                                                                                                                                                                                                                                                                                                                                                                                                                                                                                                                                                                                                                                                                                                                                                                                                                                                                                                                                                                                                                                                                                                                                                                                                                                                                                                                                                                                                                                                                                                                                                                                                                                               | 2-Nitrophloroglucinol
$(C_6H_5NO_5)$ |
| $C_{10}H_{11}NO_5$                                     | 224.0564                              | 224.0565                           | H 3 C 0 CH 3                                                                 | 2-Nitrophloroglucinol
(CaHaNOs)